# Interannual variability of summertime cross-isobath exchanges in the northern South China Sea: ENSO and riverine influences

5  **Yunping Song[1,2], Yuxin Lin[1], Peng Zhan[1], Zhiqiang Liu[1,3], Zhongya Cai[4,5]**

[1]Department of Ocean Science and Engineering, Southern University of Science and Technology, Shenzhen, China

[2]Institute for Ocean Engineering, Shenzhen International Graduate School, Tsinghua University, Shenzhen, China

10  [3]Center for Complex Flows and Soft Matter Research, Southern University of Science and Technology, Shenzhen, China

[4]State Key Laboratory of Internet of Things for Smart City, Department of Ocean Science and Technology, University of Macau, Macau, China

[5]Center for Ocean Research in Hong Kong and Macau, Hong Kong, China

*Corresponding to*: Zhiqiang Liu (liuzq@sustech.edu.cn); Zhongya Cai (zycai@um.edu.mo)

**Abstract.** This study, combining long-term observations and high-resolution simulations, investigates the interannual variability of summer shelf circulation in the Northern South China Sea (NSCS) from 2000 to 2022. We elucidate the responses of NSCS shelf circulation to ENSO and Pearl River Estuary (PRE) freshwater runoff, revealing distinct spatial and mechanistic signatures. During El Niño years, a pronounced sea level anomaly dipole forms between the central and southern South China Sea, intensifying northward geostrophic currents in the southern basin and modulating Kuroshio intrusions. Simultaneously, an amplified PRE plume extends eastward to the 100 m isobath, markedly reducing nearshore salinity. Analysis of depth-integrated vorticity equations indicates that the pressure gradient force — driven by the joint effect of baroclinicity and bottom relief (JEBAR) and bottom pressure gradients — governs NSCS circulation variability. In coastal regions, cross-isobath velocity anomalies are primarily controlled by bottom stress curl and nonlinear vorticity advection, whereas JEBAR dominates offshore dynamics beyond the 100 m isobath. During El Niño summers, bottom density anomalies generate positive cross-isobath velocity anomalies through JEBAR, partially offset by negative anomalies from altered vertical stratification, sustaining a meandering shelf current. These results highlight the interplay of regional and remote forcings, advancing understanding of NSCS hydrographic dynamics.

## 1 Introduction

The Northern South China Sea (NSCS), particularly the shelf region offshore of the Pearl River Estuary (PRE), exhibits a complex circulation system shaped by monsoonal forcing, Kuroshio intrusion, and intricate topography (Fang et al., 1998; Xue et al., 2004; Nan et al., 2015). During summer, prevailing southwesterly monsoon winds drive intense coastal upwelling, significantly modulating the hydrography of the NSCS shelf. In addition to wind-driven Ekman transport, cross-isobath exchanges are strongly influenced by topographic effects, which mainly function over the concave shelf where the isobaths show spatial irregularity (Liu et al., 2020).

The Luzon Strait, to the east of the NSCS shelf and between Taiwan Island and the Philippines (Fig. 1), serves as the primary conduit for water exchange between the NSCS and the Pacific Ocean (Zhu et al., 2019). The joint effect of baroclinicity and bottom relief (JEBAR) plays an important role comparable to wind stress in steering shelf and slope circulation, significantly influencing Kuroshio path variability near the Taiwan Strait (Oey et al., 2010, 2014). Recent studies also highlight JEBAR's critical role in cross-shelf transport across diverse shelf seas. For instance, Lin & Gan (2024) demonstrated that in the NSCS, the depth-averaged along-isobath pressure gradient is primarily driven by JEBAR, where baroclinicity dominates over topographic gradients. Tidal amplification of JEBAR further enhances the westward pressure gradient force from the outer to inner shelves east of Hainan. Similarly, Renkl et al. (2025) identified JEBAR as the dominant factor in mean dynamic topography differences on the Scotian Shelf, Canada, where it balances offshore wind-driven Ekman transport with onshore flow, sustaining coastal upwelling. Closer to the PRE, the shelf circulation takes the form of a coastal current, with the freshwater plume displaying pronounced seasonal variability. During summer, this plume extends eastward under the influence of southwesterly winds, modulating stratification and nearshore dynamics (Chen et al., 2001).

Previous studies of NSCS shelf circulation have primarily examined seasonal patterns and their wind-driven dynamics (Hu, 2000). Short-term summer variability in the shelf current has also been explored: Geng et al. (2024) demonstrated that tides modulate sea surface height, influencing shelf pressure gradients, while Liu et al. (2020) showed that cross-isobath exchanges during upwelling and downwelling winds are sensitive to along-shelf pressure gradients shaped by bathymetric irregularity. The intensity of this upwelling exhibits notable interannual variability, often linked to the El

Niño–Southern Oscillation (ENSO) (Shu et al., 2018). For example, during the summer of 1998—an El Niño year—enhanced alongshore wind stress substantially intensified coastal upwelling (Jing et al., 2011). Despite these advances, the interannual variability of summer shelf circulation remains poorly constrained, particularly in quantifying the depth structure of cross-isobath transport and attributing it to distinct processes. While expanded observational datasets, satellite products, and high-resolution modeling (Hong & Wang, 2008; Shu et al., 2011; Zu et al., 2020) have refined our understanding of NSCS dynamics, the combined effects of ENSO, PRE runoff, and regional current meandering have rarely been assessed within a unified, shelf-wide framework that computes cross-isobath transport and partitions the pressure-gradient forcing (e.g., JEBAR vs. bottom-related terms).

This study investigates the interannual variability of NSCS shelf circulation during summer from 2000 to 2022, using long-term observations and numerical modeling. We examine how ENSO modulates regional atmospheric and oceanic forcing, how the PRE runoff controls freshwater plume behavior, and how vorticity-related processes—including JEBAR, bottom stress curl, and nonlinear vorticity advection—govern cross-isobath exchanges and meandering shelf currents. Our study also provides a quantitative attribution of ENSO and river runoff impacts on cross-shelf transport in the NSCS, by explicitly computing depth-resolved cross-isobath transport (positive onshore) and decomposing the pressure-gradient/vorticity terms (JEBAR, bottom stress curl, nonlinear relative vorticity advection). By integrating these processes, our study provides new insight into how regional and remote forcings interact to shape NSCS shelf circulation, with implications for hydrographic structure and ecosystem variability.

## 2    Methods and observations

To investigate the interannual variability of summer shelf circulation in the NSCS, we employed a high-resolution regional ocean model based on the Regional Ocean Modeling System (ROMS; Shchepetkin & McWilliams, 2005). The model domain spans 105°–121° E and 16°–26° N, covering the NSCS shelf and adjacent basins. The model uses a curvilinear orthogonal Arakawa-C grid, with horizontal resolution ranging from 1 to 3 km across 800×400 grid points, enabling fine-scale representation of coastal and shelf processes. Vertically, the water column is discretized into 60

terrain-following *s*-coordinate layers, with enhanced resolution near the surface and bottom to better resolve boundary layer dynamics (Song & Haidvogel, 1994). Turbulent mixing is parameterized using the Mellor–Yamada 2.5 turbulence closure scheme, widely validated for coastal applications (Mellor & Yamada, 1982; Gan et al., 2006; Jing et al., 2009). Horizontal and vertical momentum advection are calculated using a third-order upstream and a fourth-order centered scheme, respectively, minimizing numerical diffusion while preserving accuracy. Transport of temperature and salinity employs the multidimensional positive-definite advection transport algorithm (MPDATA; Smolarkiewicz, 1984).

Open boundary conditions incorporate both tidal and subtidal forcing (Liu and Gan, 2020). Tidal elevation and velocity fields are constructed from nine principal constituents—semidiurnal ($M_2$, $S_2$, $K_2$, $N_2$), diurnal ($K_1$, $O_1$, $P_1$, $Q_1$), and the nonlinear $M_4$ component arising from $M_2$ interactions over the shallow PRE shelf (Mao et al., 2004). Harmonic constants are derived from satellite-based sea level anomaly (SLA) data processed by the Oregon Tide Inverse Software (Egbert & Erofeeva, 2002), following the approach of Zu et al. (2008). Subtidal elevation, currents, and hydrographic properties are obtained from the global Hybrid Coordinate Ocean Model (HYCOM), providing consistent large-scale boundary conditions. River runoff data for the PRE are obtained from the Chinese Ministry of Water Resources, capturing both seasonal and interannual fluctuations. Atmospheric forcing fields—including wind stress, heat fluxes, and surface pressure—are sourced from the fifth-generation ECMWF Reanalysis (ERA5), which offers high temporal and spatial resolution suitable for analyzing ENSO-related variability.

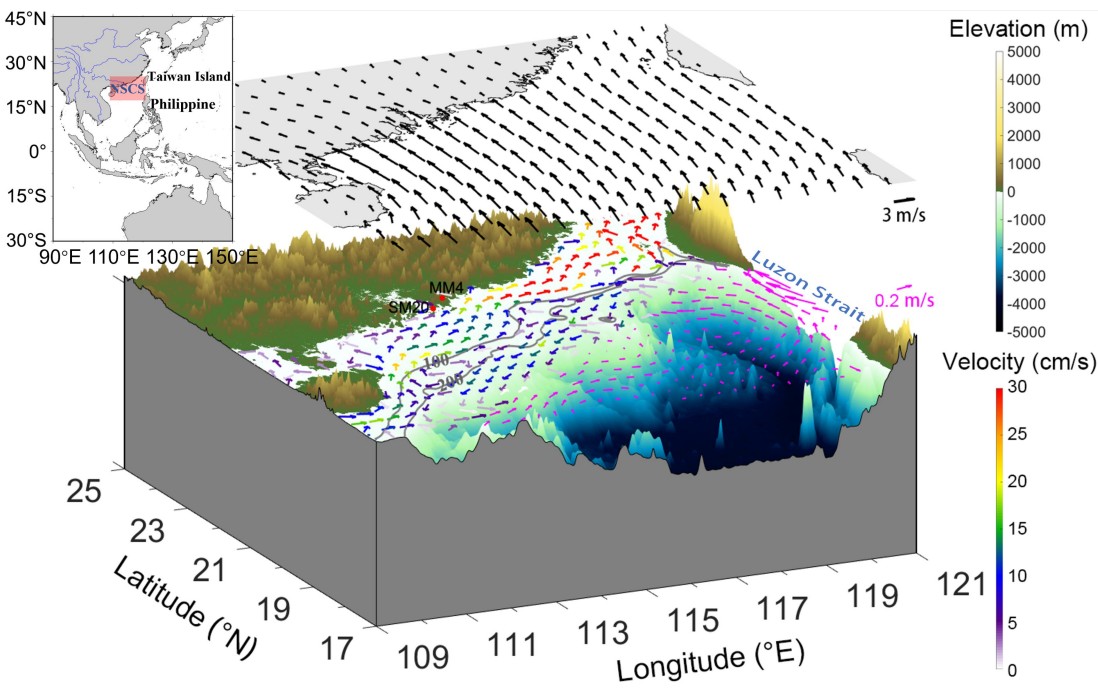

**Figure 1.** Overview map of the northern South China Sea. Model-simulated shelf currents (colored arrows) within the model domain and satellite-derived geostrophic currents (pink arrows) outside the scope of the model simulation represent summer averages from 2000 to 2022. The upper panel shows the corresponding climatological wind field. Red markers indicate the observational sites used for model validation in Fig. 2. The upper left insert shows the location of the northern South China Sea.

The model simulation spans the period 1994–2022, with analysis focused on summer months (June–August) from 2000 to 2022. The initial six-year period (1994–1999) is excluded to allow for model spin-up and dynamic equilibration. Model performance has been rigorously validated in previous NSCS studies, demonstrating skill in reproducing sea level, currents, temperature, and salinity (Liu et al., 2020; Deng et al., 2022; Song et al., 2024). Summer SST anomalies (2000-2022) at monitoring stations (Fig. 1) show strong agreement with observations (CC=0.74, RMSE=0.65 ℃; Fig. 2a). Notably, SSS anomalies achieve higher accuracy (CC=0.83, RMSE=1.25 psu; Fig. 2b), particularly in capturing the PRE plume dynamics at interannual scales. The model also skillfully reproduces SLA variability from the EU Copernicus Marine Service (CMEMS) seaward of the 200-m isobath (CC=0.72, RMSE=2.27 cm; Fig. 2c), effectively resolving ENSO-related and regionally forced variabilities. Moreover, to assess currents where observations exist, we compare our simulation with summertime high-resolution ADCP records over the shelf off the Pearl River Estuary reported by Liu

et al. (2020) for the same site and period, and we examine residual (de-tided) sea level at the Waglan Island tide gauge. The simulated along- and cross-shelf velocities reproduce the observed fluctuations with realistic amplitude and phase, and the residual sea level shows consistent subtidal variability (Fig. A1). These validation metrics demonstrate that the model reliably captures key physical processes governing shelf circulation, cross-isobath transport, and vorticity dynamics in the NSCS, providing a robust foundation for the analysis presented in this study.

We use Multivariate Empirical Orthogonal Function (MVEOF) to extract the dominant coupled spatio-temporal modes shared by multiple, related variables (Dawson, 2016; Liang et al., 2018). The method extends conventional EOF by forming a joint covariance structure that includes cross-covariances among the selected fields (e.g., sea-surface height, temperature, and velocity), thereby identifying patterns that maximize joint variance across variables. Each mode is paired with a principal-component (MVPC) time series that describes its temporal evolution, and with variable-wise spatial maps (from regressing the MVPC back onto each field) that show how the variables co-vary in space. These maps are derived from the same MVPC time series. The spatial patterns across variables reflect a consistent, objectively derived co-varying structure.

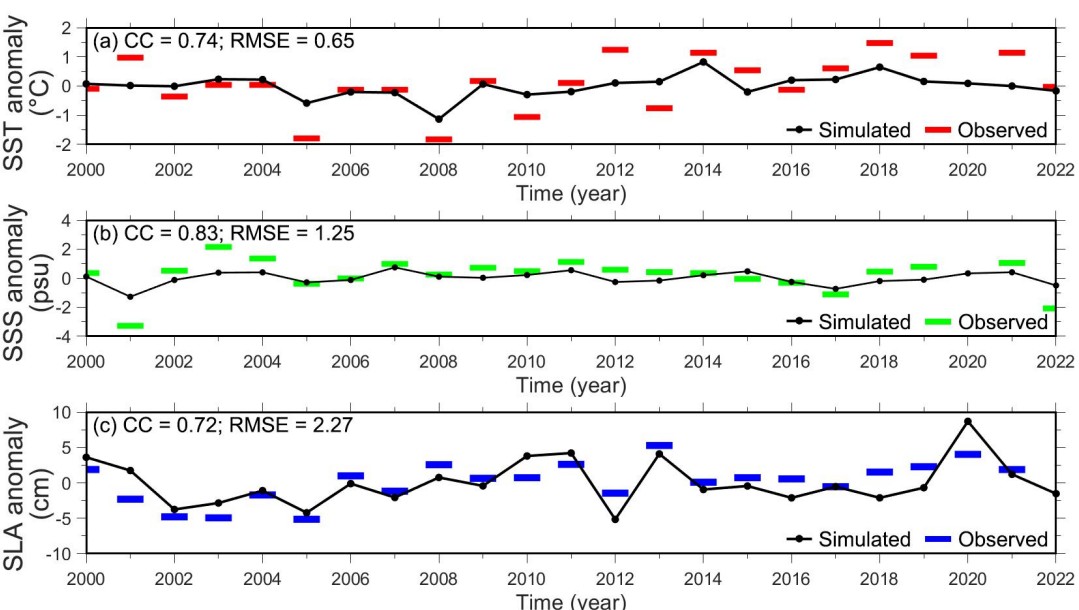

**Figure 2.** Temporal variations of observed and simulated anomalies during summer (2000–2022): (a) sea surface temperature (SST, °C) at site SM20; (b) sea surface salinity (SSS, psu) at site MM4; (c) sea level anomaly (SLA; cm) averaged seaward of the 200 m isobath within the model domain.

# 3    Results

## 3.1 Interannual variability of dynamic factors influencing NSCS shelf circulation in summer

To assess the interannual variability of summer shelf circulation and associated hydrographic properties in the NSCS, we analyzed a suite of dynamic factors encompassing atmospheric and oceanic processes, including evaporation minus precipitation (E-P), air temperature, wind stress, SST, SLA, and SSS, all aggregated at annual (June–August) resolution. To extract coherent spatiotemporal patterns, we applied multivariate empirical orthogonal function (MVEOF) analysis using the eofs Python library (Dawson, 2016). The covariance matrix was computed from the anomaly fields (time-mean removed) and normalized by N−1, where N is the number of temporal samples. This normalization provides an estimate of the population covariance. This approach identifies dominant modes of variability and their linkages to large-scale climate drivers, particularly ENSO, as a key indicator of regional interannual variability.

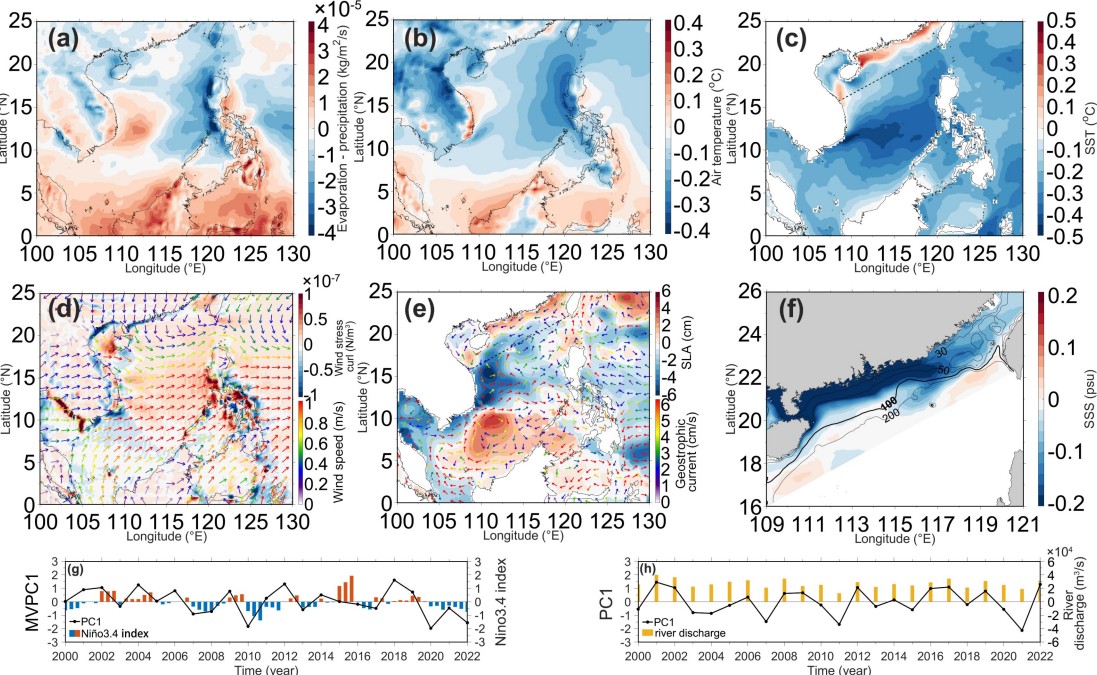

**Figure 3.** Leading mode (Mode 1) of summer-averaged climatic and oceanic variables from multivariate EOF (MVEOF): (a) evaporation minus precipitation (kg m⁻² s⁻¹); (b) air temperature at 2 m height (°C); (c) sea surface temperature (°C); (d) sea surface wind stress curl (N m⁻³; shown by the background color) and wind speed (m s⁻¹;

denoted by the arrows); and (e) sea level anomaly (SLA; cm) with geostrophic currents (cm s$^{-1}$; denoted by the arrows), all representing the spatial pattern of MVEOF Mode 1; (f) sea surface salinity (SSS; psu) spatial mode from univariate EOF; (g) principal component time series of MVEOF Mode 1 (MVPC1, black line) with Niño 3.4 index (colored bars), and (h) principal component time series of SSS EOF Mode 1 (PC1, black line) with superimposed PRE summer runoff. Anomalies are relative to summer climatology, and vectors are scaled for clarity.

The leading mode (Mode 1) and its corresponding MVPC1 are shown in Fig. 3, highlighting spatial patterns and their temporal evolution. Mode 1 explains 60.60% of the total variance, with MVPC1 exhibiting a correlation coefficient of 0.62 with the Niño 3.4 index. The spatial structures associated with Mode 1 reveal contrasting responses during years with positive and negative MVPC1 phases, which, in general, correspond to El Niño and La Niña years, respectively. In positive MVPC1 years, a weakened Walker circulation displaces atmospheric pressure centers and trade-wind zones southward, while negative phases strengthen the circulation and shift them northward. This pattern during the positive MVPC1 years establishes positive wind stress curl over the South China Sea (SCS) basin, and a high-pressure anomaly over the southern SCS and generates a prominent SLA dipole across the SCS, featuring positive SLA anomalies south of 10° N and negative anomalies north of 10° N (Fig. 3e). Simultaneously, under the control of the anomalous atmospheric high-pressure (low-pressure), the clear (cloudy) sky during positive (negative) MVPC1 phases enhances (suppresses) evaporation and raises (lowers) air temperatures in the southern SCS. On the contrary, the air temperature and SST decrease (increase) in the mid-basin of the SCS during positive (negative) MVPC1 years.

Over the NSCS shelf, E-P (Fig. 3a) and air temperature (Fig. 3b) tend to decrease (increase) during positive (negative) MVPC1 years, leading to cooler (warmer) SST seaward of the 200 m isobath (Fig. 3c). In contrast, regions landward of this isobath experience the opposite trend with warmer (cooler) SST in positive (negative) phases, due to substantial freshwater input from the PRE and intensified rainfall over the Southern Chinese mainland. During El Niño summers (positive MVPC1 phases), northeasterly wind stress anomalies (Fig. 3d) develop over the NSCS, acting against the prevailing southwesterly monsoon and thereby suppressing coastal upwelling and contributing to SST increases nearshore. At the same time, the Kuroshio intrusion through the Luzon Strait intensifies, strengthening the inflow of cooler waters during the positive phase (Fig. 3e). The strengthened geostrophic currents in the southern SCS (Wang et al., 2020) are linked to anticyclonic (cyclonic) wind stress curl anomalies

in the central and western SCS basin and offshore Vietnam (Fig. 3e).

In contrast, interannual SSS variability in the NSCS is primarily governed by PRE runoff rather than directly linked to ENSO (Fig. 3f, h). A separate EOF analysis of SSS shows that Mode 1 accounts for 60.55% of the total variance, with PC1 closely tracking observed PRE runoff (CC = 0.83; Fig. 3h).

PC1 of SSS EOF analysis (Fig. 3h) corresponds to PRE runoff variability, with positive phases reflecting large-runoff years (Table 2) and negative phases low-runoff years. During large-runoff summers, SSS decreases markedly as the freshwater plume extends eastward to approximately the 100 m isobath near 115° E (Fig. 3f). This indicates that regions inshore of the 100 m isobath are more sensitive to riverine input, as the plume readily spreads across the shelf. These results emphasize the

dominant role of riverine freshwater in modulating nearshore salinity and highlight the need to consider local riverine forcing alongside broader climatic influences.

The strong correlations between MVPC1 and the Niño 3.4 index (Fig. 3g), and between PC1 and PRE runoff (Fig. 3h), motivated analyses of shelf currents anomalies for the years listed in Tables 1 and 2, which respectively represent the conditions in the positive and negative MVPC1 and PC1 years.

However, the relationship between summer PRE runoff and ENSO remains controversial. For instance, Gu et al. (2017) and Ouyang et al. (2014) reported increased precipitation and flows in the Pearl River during La Niña compared to El Niño periods, whereas Wang et al. (2018) and Xu et al. (2007) noted higher precipitation in South China during El Niño and lower precipitation during La Niña. To isolate the riverine plume's influence on shelf currents, we selected large-runoff years from El Niño periods

and low-runoff years from La Niña periods, denoted by bold labels in Table 1 and summarized in Table 2, to disentangle runoff effects. Any year with a summer-averaged river discharge exceeding 26,000 m³ s⁻¹ (70th percentile for these years) is classified as a high-runoff year, while sensitivity to the choice of percentile is shown in Fig. S1 of the Supplement.

**Table 1.** The El Niño (positive MVPC1) and La Niña (negative MVPC1) years. Shelf circulation anomalies in El Niño years with low runoff and in La Niña years with high runoff are generally opposite to those observed in high runoff El Niño years; the former cases, listed as non-bold entries in Table 1, were therefore excluded from subsequent analyses.

| | El Niño | La Niña |
|---|---|---|
| **Years** | 2000, **2001, 2002,** 2004, 2005, **2006, 2009,** 2011, **2012, 2014,** 2015, 2018, **2019** | **2003, 2007,** 2008, **2010, 2013,** 2016, 2017, **2020, 2021,** 2022 |

Table 2. The large runoff (positive PC1) and low runoff (negative PC1) years

| | Large runoff | Low runoff |
|---|---|---|
| Years | 2001, 2002, 2006, 2009, 2012, 2014, 2019 | 2003, 2007, 2010, 2013, 2020, 2021 |

**3.2 Climatic characteristics of NSCS shelf circulation in summer**

This section investigates the climatological characteristics (2000 – 2022) of summer NSCS shelf circulation, where the mean circulation is dominated by southwesterly monsoon winds (Fig. 1). These winds drive prominent upwelling currents in coastal waters shallower than 100 m (Fig. 4a). Conversely, offshore regions beyond the shelf break exhibit southwestward transport, primarily due to Kuroshio intrusion through the Luzon Strait, consistent with prior studies (e.g., Li et al., 2018). Interannual wind anomalies further modulate this circulation: during positive MVPC1 years (El Niño conditions; see Sect. 3.1), northeastward wind stress anomalies arise (Fig. 3e) and reflect ENSO's climatic influence on NSCS circulation.

To systematically characterize interannual anomalies in shelf currents and hydrographic properties, we implemented a two-stage regression – composite framework. First, we regressed the anomaly field of each variable on its corresponding MVPC1 or PC1 index (Fig. 3g, h). Second, we composited the resulting regression patterns during positive MVPC1 or PC1 phases (Tables 1 – 2) to emphasize persistent spatial structures. Together, these steps isolate MVPC1 or PC1-linked variability and enhance recurrent spatial signals while suppressing short-term noise. Although the regression patterns resemble the simple conditional-composite maps (Fig. S2 of the Supplement), regression yields directly interpretable amplitudes and enforces strict antisymmetry between positive and negative phases.

To explore the cross-isobath water exchanges associated with the upwelling circulation, we decomposed the depth-averaged velocity field into along-isobath (positive values indicate flow with shallower waters on the right) and cross-isobath (positive values indicate flow toward shallower waters) components. The summer-averaged cross-isobath velocity indicates net onshore transport across most of the NSCS shelf (Fig. 4b), except where meandering isobaths cause topographic steering to locally alter flow direction. Specifically, using the

convention "positive = onshore", summertime cross-isobath flow over the NSCS shelf is on the order of $O(10^{-2}\,\mathrm{m\,s^{-1}})$ on average, with substantial spatial variability and localized peaks that can briefly exceed $O(10^{-1}\,\mathrm{m\,s^{-1}})$ near dynamical hotspots such as the Taiwan Shoal (Fig. 4b). During El Niño summers, interactions between northeastward wind stress anomalies and shelf topography generate a distinct southwestward meandering current anomaly across the central NSCS shelf (Fig. 4c, d). This pattern highlights the nonlinear interplay between wind forcing and bathymetric gradients in driving interannual shelf circulation variability.

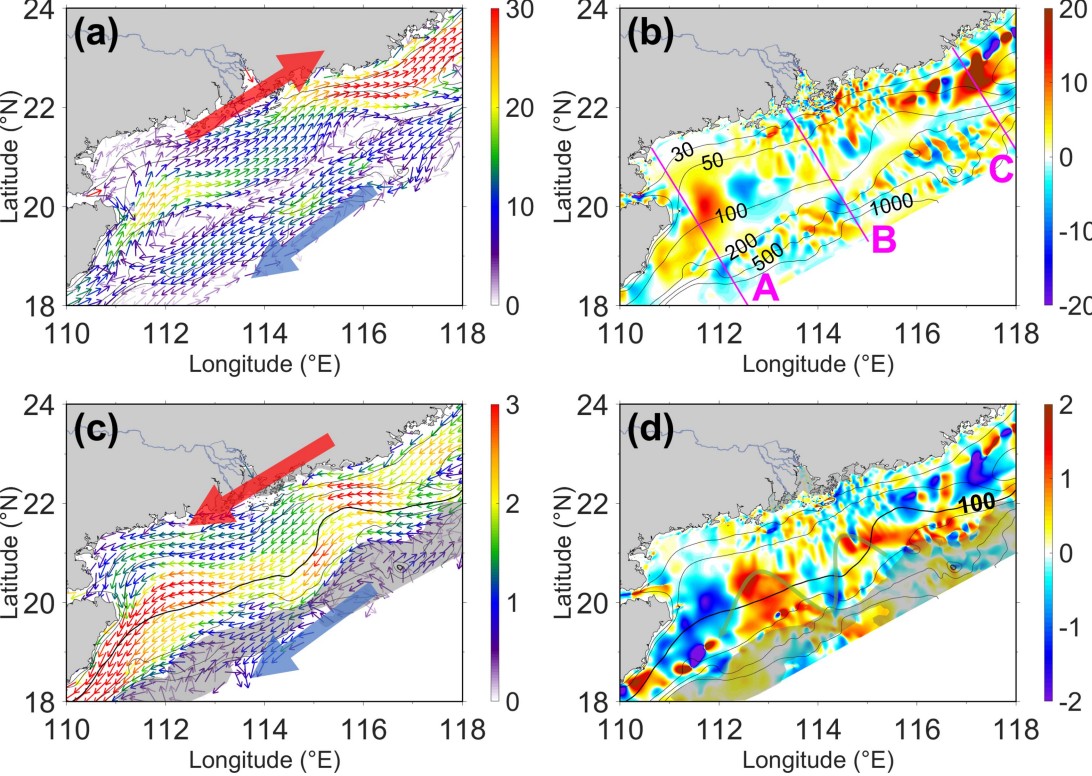

**Figure 4.** Depth-averaged circulation and associated anomalies in the northern South China Sea: (a) mean velocity vectors (cm s⁻¹) and (b) mean cross-isobath velocity (cm s⁻¹) averaged over summers from 2000 to 2022, where positive cross-isobath velocity indicates flow from deeper to shallower waters; (c) regression map of velocity vector anomalies (cm s⁻¹), and (d) regression map of cross-isobath velocity anomalies (cm s⁻¹) during positive MVPC1 years (Table 1). Shaded areas (mainly over the continental slope) in (c) and (d) denote regions where the 90% confidence level is not met; this convention is followed in subsequent regression maps, based on a two-tailed t-test using the estimated standard deviation and sample degrees of freedom. For the simple linear regression, the degrees of freedom are N-2 (N is the number of observations), assuming independent observations. As temporal autocorrelation may reduce the effective number of independent samples, the reported regions with 90% confidence level may include uncertainties. The two-stage regression approach is detailed in Section 3.2. Positive values in (c) and (d) indicate flow toward shallower waters.

To further assess hydrographic characteristics under varying climatic conditions, we conducted a hydrographic analysis along three cross-shelf transects (marked in pink in Fig. 4b), with corresponding temperature and salinity sections shown in Fig. 5 during El Niño or large runoff years. During positive MVPC1 years—coinciding with weakened monsoonal forcing—all three transects exhibit elevated temperatures across the shelf, while colder, Kuroshio-originated waters occupy the continental slope (Fig. 5a–c). Along Transect A, east of Hainan Island, anticyclonic wind stress anomalies enhance the surface expansion of the warm PRE plume, reinforcing the coastal current. In contrast, Transect C, located further east and exposed to cyclonic anomalies, displays suppressed vertical mixing and elevated SST, indicative of a regime that is unfavorable for upwelling circulations.

The influence of buoyant freshwater input becomes especially pronounced during years with large PRE runoff. Under prevailing southwesterly winds, fresher water spreads widely across the shelf, most prominently at Transect C (Fig. 5d–f). Concurrently, saltier Kuroshio waters are observed over the slope, particularly at Transect A, contributing to enhanced stratification. This establishes a stratified system driven by deeper saline inflows and wind-induced divergence. In addition, the east-west contrast of this system manifests that the water exchanges in the water column are prohibited over the eastern shelf due to surface freshening, while upwelling-favorable conditions persist over the western shelf. It is also noteworthy from this figure that landward of the 100-m isobath (excluding the immediate nearshore), the first baroclinic Rossby radius ranges from a few to ~10 km. The radius is given by $R_o = \sqrt{g'H}/f$, where $H$ is the water depth and $f$ represents the Coriolis parameter. The reduced gravity is defined as $g' = \Delta\rho/\rho$, with $\Delta\rho$ the density difference between the upper and lower uniform layers, and $\rho$ the domain-averaged density as reference density. This scale exceeds the model grid spacing yet remains an order of magnitude smaller than the shelf width, supporting the validity of the climatic scales adopted in this study. At the climatic scales in this study, the Rossby number is small, so nonlinear advection remains dynamically secondary in the depth-averaged momentum balance, and the large-scale shelf circulation is close to geostrophic, particularly in the cross-shelf direction.

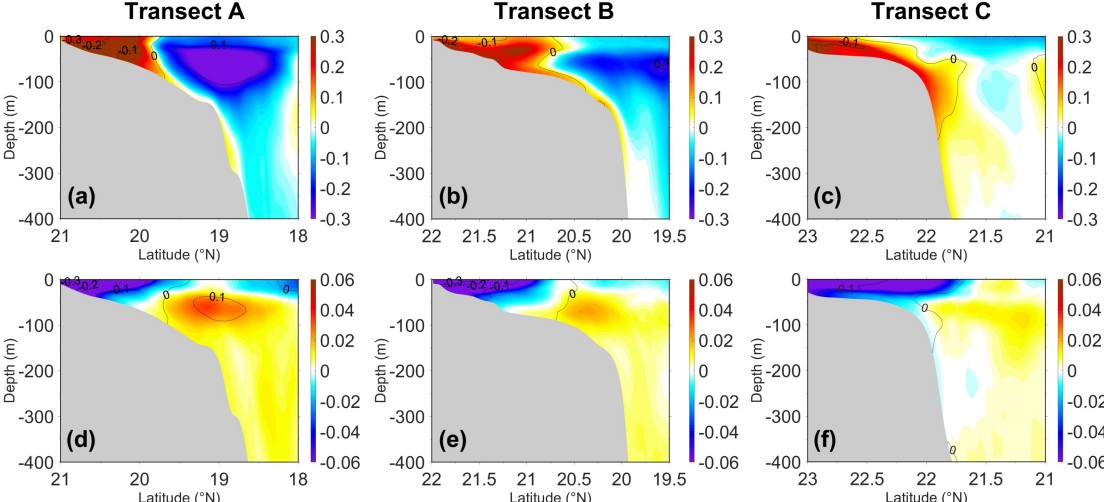

**Figure 5.** Regression maps of hydrographic anomalies along three cross-shelf transects (locations shown in Fig. 4b): (a–c) temperature anomaly profiles (°C) during the summer of positive MVPC1 years (Table 1) at Transects A, B, and C, respectively; (d–f) salinity anomaly profiles (psu) during positive PC1 years (Table 2) at the same transects. The isolines depict the regressed density anomalies for the corresponding scenarios. The two-stage regression approach is detailed in Section 3.2. To maintain the visual clarity of the anomaly cores, which are confirmed to exceed the 90% confidence level, significance shading is omitted from the profiles, and this approach is also applied in Fig. 7 and Fig. 9.

## 4 Mechanisms altering the interannual variability of cross-isobath transport

To investigate the mechanism altering the interannual variability of cross-isobath transport, we adopt the depth-averaged vorticity equation under conditions of low Rossby and Ekman numbers, appropriate for the interannual timescale considered, as outlined in Liu et al. (2020):

$$PGF_{y*} = JEBAR + PGF_{y*}^b \qquad (\;1\;)$$

$$PGF_{y*}^b = \overbrace{\frac{1}{H_{x*}}\overline{\nabla}\times\left(\frac{\tau_b}{\rho_0}\right)}^{BSC} + \overbrace{\frac{1}{H_{x*}}\overline{\nabla}\times\left(-\frac{\tau_s}{\rho_0}\right)}^{SSC} + \overbrace{\frac{1}{H_{x*}}J(\psi,\xi)}^{RVA} + \overbrace{\left(\frac{\|\vec{v}\|^2}{2}\right)_{y*}}^{GMF} \qquad (\;2\;)$$

In these equations, the subscripts $(x*)$ and $(y*)$ represent the cross-isobath and along-isobath derivatives, respectively. The $x*$ is positive onshore and $y*$ is in along-isobath direction and positive $y*$ stands for the direction with deeper waters on the left-hand-side. In the terms $PGF_{y*}$ and $PGF_{y*}^b$, the subscript $y*$ represents the along-isobath direction rather than derivative. $H$ is the water depth; $\tau_s$ and $\tau_b$ represent sea surface wind stress and bottom shear stress, respectively; $J$ is the Jacobi operator; $\psi$ is the transport streamfunction, $\xi$ is the relative vorticity, and $\vec{v}(\overline{u},\ \overline{v})$ denotes

the depth-averaged velocity, respectively. According to Mertz & Wright (1992), the

depth-averaged along-isobath pressure gradient force ($PGF_{y*}$) in the water column can be

decomposed into two terms: JEBAR and bottom along-isobath pressure gradient force ($PGF_{y*}^b$).

The $PGF_{y*}^b$ can be further decomposed into bottom stress curl (*BSC*), surface stress curl (*SSC*), the

nonlinear advection of relative vorticity (*RVA*), and the gradient of momentum flux (*GMF*) terms (Liu

and Gan, 2015). All the composite terms above are normalized by the Coriolis parameter to further

assess their contribution to the cross-isobath velocity anomaly, and the results are presented in Fig. 6.

The cross-isobath velocity anomaly contributed by the term GMF during positive MVPC1 years is

negligible compared with the other terms (Fig. S3), thus it is excluded from the following discussion.

The cross-isobath velocity anomaly associated with JEBAR (Fig. 6a) predominantly governs

transport from the vicinity of the 100 m isobath to the SCS basin, whereas the influence of $PGF_{y*}^b$ (Fig.

6b) is largely concentrated over the shelf inshore of the 100 m isobath, as also shown in Liu et al.

(2020). More specifically, the spatial distribution of water exchange induced by the nonlinear advection

of relative vorticity (RVA; Fig. 6c) forms the principal structure of the $PGF_{y*}^b$ contribution across the

shelf, particularly between the 30 m and 100 m isobaths. For instance, over the shelf east of Hainan

Island, the strong resemblance between Fig. 6b and c highlights RVA as the dominant contributor in

this region. Thus, while nonlinear advection is secondary in the large-scale depth-averaged momentum

balance, it can still emerge as the leading contributor to the $PGF_{y*}^b$ and thereby shape the detailed

structure of cross-isobath velocity anomalies over the coastal band. The BSC term (Fig. 6d) is notably

influenced by complex bathymetry, such as near the Taiwan Shoal, where it generates offshore flow

near the 50 m isobath and pronounced onshore transport inshore of the 30 m isobath during positive

MVPC1 years. Overall, the meandering cross-isobath velocity anomaly (Fig. 4d) results from the

combined influence of JEBAR, which controls offshore exchange beyond the 100 m isobath, and

$PGF_{y*}^b$, which plays a central role in shaping the hydrodynamics of the coastal seas.

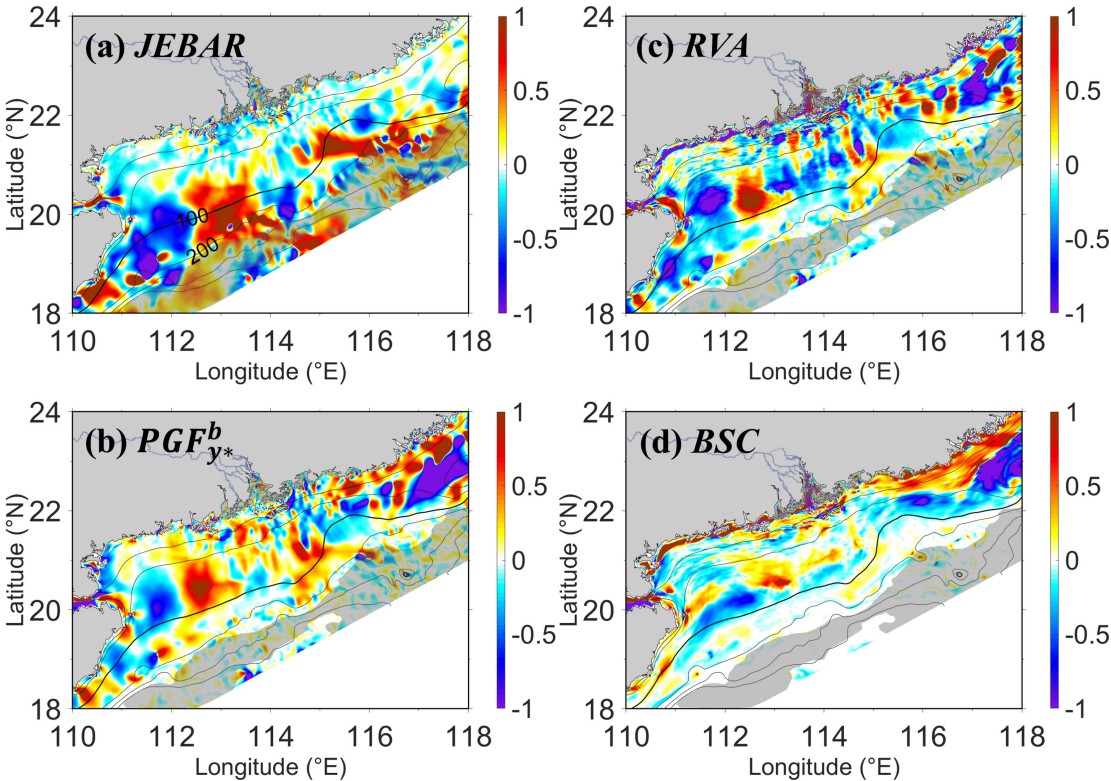

**Figure 6.** Regression maps of horizontal cross-isobath velocity anomalies (cm s⁻¹) during the summer of positive MVPC1 years (Table 1), attributed to different dynamic components: (a) joint effect of baroclinicity and bottom relief (JEBAR); (b) bottom pressure gradient force ($PGF_{y*}^b$); (c) nonlinear relative vorticity advection (RVA); and (d) bottom stress curl (BSC). The shaded areas indicate regions that fail to attain the 90% confidence level. For the simple linear regression, the degrees of freedom are N-2 (N is the number of observations), assuming independent observations. As temporal autocorrelation may reduce the effective number of independent samples, the reported regions with 90% confidence level may include uncertainties. The two-stage regression approach is detailed in Section 3.2. Positive values in the figures indicate flow toward shallower waters. The contribution of the GMF term is shown in Fig. S3 in the Supplement.

Given that the 100 m isobath marks a critical boundary separating shelf dynamics from those of the deeper offshore region, the circulation patterns along this isobath are important to reveal the underlying mechanisms of cross-isobath exchange. During positive MVPC1 years, the regressed cross-isobath velocity anomaly over the 100 m isobath (Fig. 7a) reveals a vertically coherent offshore transport from the surface to the bottom, with notable intensification near the widened shelf around 115° E and over the shelf east of 118° E and west of 113° E. Despite the overall coherence of the velocity structure, the hydrographic properties associated with these currents show distinct characteristics. East of 115° E over the 100 m isobath, the flow is predominantly accompanied by

warmer, buoyant freshwater discharged from the PRE (Fig. 7b), while west of 115° E, colder

Kuroshio-intruded water dominates the offshore transport, particularly during El Niño summers, as also

previously shown in Fig. 3. From a salinity perspective, the PRE plume exerts a significantly stronger

influence to the east of 115° E, especially in the surface and middle layers (Fig. 7c). This spatial

contrast persists even during years of elevated freshwater runoffs, particularly under positive PC1

conditions. These findings highlight the essential role of the PRE plume in modulating the salinity

structure of the NSCS, with its influence being most pronounced on the eastern shelf, where it interacts

with offshore currents and contributes to broader shelf-basin exchange processes through modulating,

for example, the JEBAR.

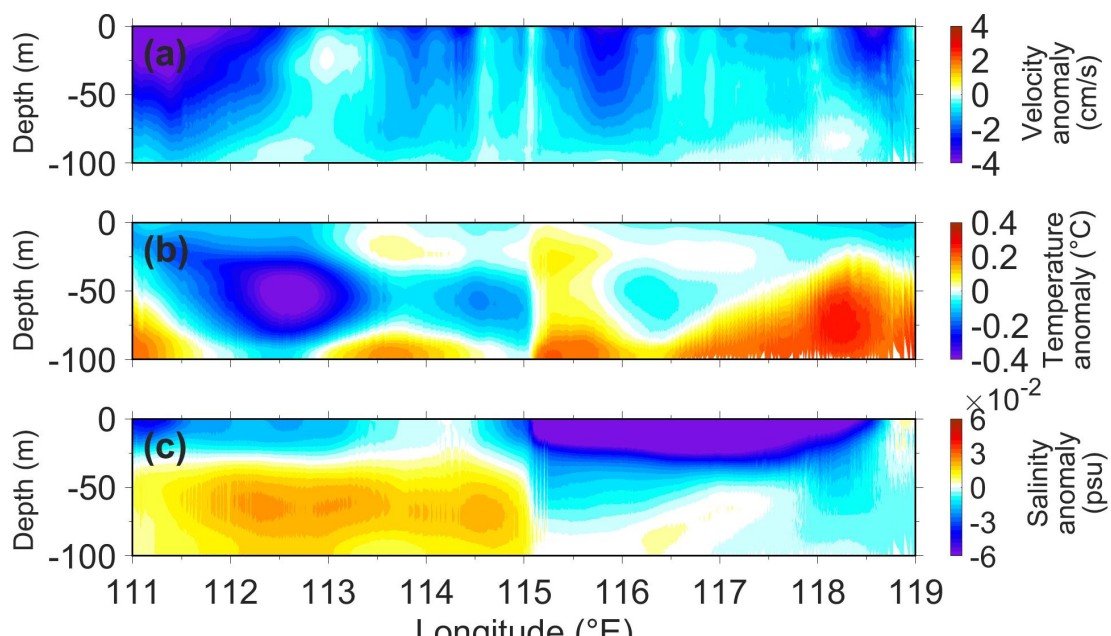

**Figure 7.** Regression maps of hydrographic anomalies along the 100 m isobath: (a) cross-isobath velocity anomaly (cm s$^{-1}$) and (b) temperature anomaly profiles (°C) during positive MVPC1 years (Table 1); (c) salinity anomaly profile (psu) during positive PC1 years (Table 2). The two-stage regression approach is detailed in Section 3.2.

Based on the previously identified interannual characteristics of hydrographic properties and

drawing from Mertz & Wright (1992), we extend the analysis of the JEBAR in Equation 3:

$$\text{JEBAR} = J\left(\frac{g}{\rho_0}\int_{-H}^{0} z\rho dz, H^{-1}\right) \tag{3}$$

Equation 3 can be explicitly rewritten in a Cartesian coordinate system:

$$\text{JEBAR} = \left(\frac{g}{\rho_0}\int_{-H}^{0} z\rho dz\right)_x \left(\frac{1}{H}\right)_y - \left(\frac{1}{H}\right)_x \left(\frac{g}{\rho_0}\int_{-H}^{0} z\rho dz\right)_y \tag{4}$$

The subscripts $(x)$ and $(y)$ in this derivative represent the derivatives in the zonal and meridional directions. Given that the water depth $H$ generally remains constant—since variations in the SLA are much smaller than the topographic depth—and that the slope current streams in the NSCS show minimal spatial displacements between positive and negative MVPC1 and PC1 years, we focus our analysis on further understanding the zonal and meridional gradients of the baroclinic component: $\frac{g}{\rho_0}\int_{-H}^{0} z\rho dz$, expressed by:

$$\frac{g}{\rho_0}\int_{-H}^{0} z\rho dz = -\frac{g}{\rho_0}\int_{-H}^{0} \frac{z^2}{2}\frac{\partial \rho}{\partial z}dz - \frac{g}{\rho_0}\frac{H^2}{2}\rho_b \tag{5}$$

The first term on the right-hand side of Equation 5 quantifies the strength of density stratification in the water column over the shelf, while the second term represents the density anomaly of the bottom seawater. Our primary focus is on the interannual variability of the cross-isobath velocities, which are shown in Fig. 5. To further investigate the anomaly of the baroclinic component, we compute it as follows:

$$\frac{g}{\rho_0}\int_{-H}^{0} z(\rho - \bar{\rho})dz = \overbrace{-\frac{g}{\rho_0}\int_{-H}^{0} \frac{z^2}{2}\frac{\partial(\rho - \bar{\rho})}{\partial z}dz}^{N'} + \overbrace{-\frac{g}{\rho_0}\frac{H^2}{2}(\rho - \bar{\rho})_b}^{P'} \tag{6}$$

The spatial distribution pattern of $\left(\frac{g}{\rho_0}\int_{-H}^{0} z\rho dz\right)'$ aligns closely with the meandering pathways of the shelf current along the 100 m isobath, as shown in Fig. 4d, particularly during positive PC1 years under both strong (Fig. 8a) and low runoff conditions (Fig. 8d). The region adjacent to the 100 m isobath consistently satisfies the 90% confidence level. The horizontal distributions of each term in Equation 6 during positive MVPC1 years are displayed in Fig. 8, with corresponding patterns for negative MVPC1 years shown in Fig. S4 of the Supplement. It is evident that the $P'$ (Fig. 8c, f) plays a dominant role in shaping the pattern of $\left(\frac{g}{\rho_0}\int_{-H}^{0} z\rho dz\right)'$. However, the negative values near $114^\circ$E, primarily sourced from $N'$ (Fig. 8b, e), are critical for cross-shelf water transport. Notably, the strength of the term $\left(\frac{g}{\rho_0}\int_{-H}^{0} z\rho dz\right)'$ is highly correlated with $P'$, the key determinant of which is the bottom density anomaly. In contrast, the variation in $N'$, which is primarily influenced by the intensity of density stratification, appears less pronounced during both positive and negative PC1 years, especially near the 100 m isobath. However, in negative MVPC1 years, this term exhibits more significant differences, as also shown in Fig. S4a, c in the Supplement. Consequently, the negative values near the 100 m isobath are more prominent when runoff from the PRE is relatively low, as seen in Fig. S4 in the

Supplement.

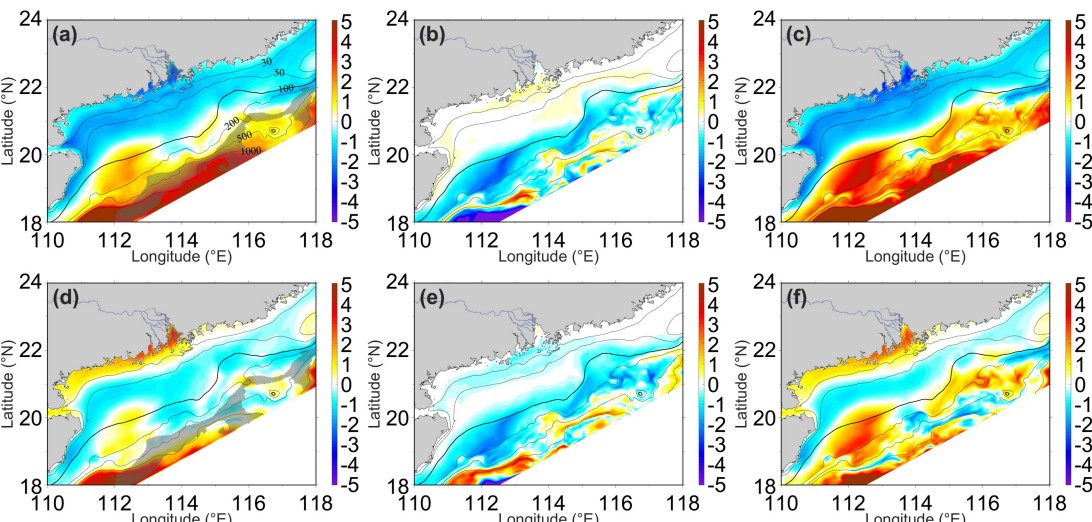

**Figure 8.** Regression maps of horizontal components (m³ s⁻²) in the JEBAR term during the summer of positive MVPC1 years, under different runoff conditions: (a–c) correspond to positive PC1 years, and (d–f) to negative PC1 years. Panels show (a, d) the full baroclinic gradient term ($\frac{g}{\rho_0}\int_{-H}^{0} z(\rho - \bar{\rho})dz$), (b, e) the contribution from vertical density stratification ($-\frac{g}{\rho_0}\int_{-H}^{0} \frac{z^2}{2}\frac{\partial(\rho-\bar{\rho})}{\partial z}dz$), and (c, f) the contribution from bottom density anomaly ($-\frac{g}{\rho_0}\frac{H^2}{2}(\rho - \bar{\rho})_b$). The shaded areas denote the approximate regions that do not pass the 90 % confidence test. For the simple linear regression, the degrees of freedom are N-2 (N is the number of observations), assuming independent observations. As temporal autocorrelation may reduce the effective number of independent samples, the reported regions with 90% confidence level may include uncertainties. The MVPC1 time series is shown in Fig. 3g, with corresponding positive-phase years listed in Table 1, while the PC1 time series is shown in Fig. 3h with its positive-phase years listed in Table 2. The two-stage regression approach is detailed in Section 3.2. The region adjacent to the 100 m isobath consistently satisfies the 90 % confidence level.

Having identified the determining factor responsible for the interannual variability of cross-isobath water transport over the 100 m isobath, we now focus on examining the density and buoyancy frequency anomaly profiles during positive MVPC1 years to elucidate the underlying formation mechanisms. The scenarios during negative MVPC1 years are presented in Fig. S5 of the Supplement. Longitudinal profiles at Transect B for different conditions are shown in Fig. S6 of the Supplement. These profiles reveal that the density anomaly patterns west of 115° E (the widened shelf) during both positive and negative PC1 years exhibit a similar structure (Fig. 9a, c). A prominent feature is the core of exceptionally high density anomaly between 112° and 113° E, strongly influenced by the southward flow of Kuroshio water. In contrast, the eastern side of the widened shelf presents a

markedly different situation. Due to the substantial influx of buoyant water from the PRE, surface density decreases during positive PC1 years (Fig. 9a), while bottom density remains largely unchanged (Fig. S5a, b of the Supplement). In essence, the term $P'$ over the 100 m isobath shows little variation regardless of whether runoff is high or low, supporting the distribution patterns observed in Fig. 8c, f.

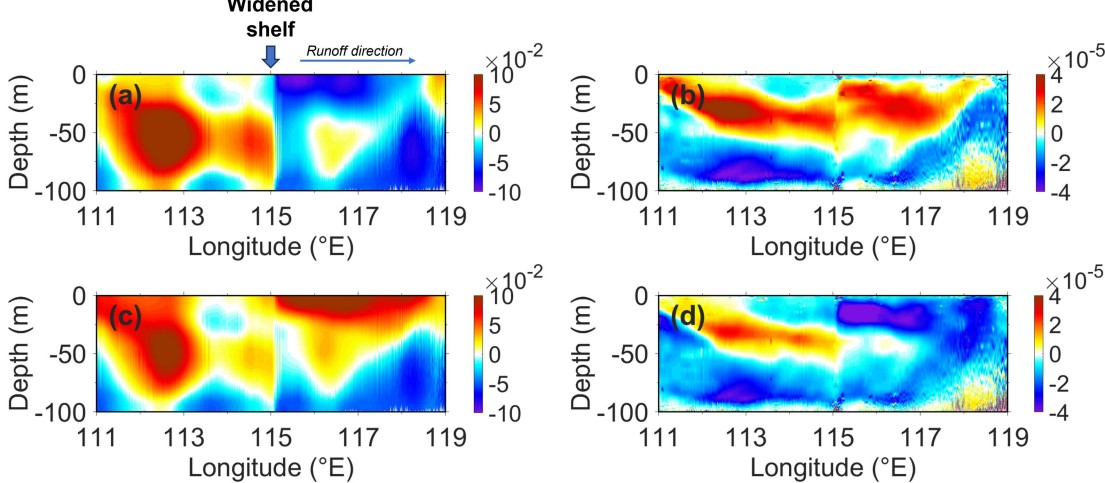

**Figure 9.** Regression maps of hydrographic structure anomalies along the 100 m isobath during the summer of positive MVPC1 years (Table 1), under varying runoff conditions: (a, b) correspond to positive PC1 years, and (c, d) to negative PC1 years (Table 2). Panels show (a, c) density anomaly profiles (kg m⁻³) and (b, d) buoyancy frequency squared anomaly profiles (s⁻²). The two-stage regression approach is detailed in Section 3.2.

The vertical profile of the buoyancy frequency anomaly during both positive and negative PC1 years further reveals distinctions between the western and eastern parts of the shelf. During years with large runoff, the density stratification in the surface and middle layers is enhanced across almost the entire 100 m isobath (Fig. 9b). In contrast, during years with low runoff, stratification is only strengthened on the western side of the widened shelf (Fig. 9d), and its intensity is lower compared to positive PC1 years. Furthermore, surface density stratification during low runoff years is noticeably weaker than in the mean state (Fig. S5c, d of the Supplement). In summary, during positive MVPC1 years, the vertical integration of the buoyancy frequency anomaly over the 100 m isobath is more pronounced in years of large runoff from the PRE than during low runoff periods. This explains the increased presence of negative values over the 100 m isobath in Fig. 8d compared to Fig. 8a, which in turn directly influences the interannual variability of the spatial distribution of JEBAR.

**5   Conclusions**

This study investigates the interannual variability of shelf circulation in the NSCS during the summer months from 2000 to 2022, utilizing long-term observational and numerical simulation data. The findings indicate that this variability is strongly influenced by ENSO-driven climate dynamics and river runoff from the PRE. Based on the MVEOF and an independent EOF analysis, the ENSO-related

mode explains approximately 60% of the large-scale circulation variance over the SCS in summer. By contrast, within the NSCS, cross-isobath transport is largely associated with river discharge, contributing up to ~60%. During El Niño years, the warm pool of the tropical Northwest Pacific shifts eastward, leading to a cooler SST and a pronounced westward Kuroshio intrusion through the Luzon Strait in the eastern SCS. This shift induces a cyclonic wind stress curl anomaly in the NSCS and an

anticyclonic wind stress curl anomaly in the southern region, resulting in a distinct dipole pattern in the SLA anomaly across the central and southern SCS basin adjacent to the NSCS. Additionally, the northward geostrophic current is intensified in the southern SCS during El Niño years. When a warmer and more extensive Pearl River plume is generated, likely due to increased terrestrial precipitation, the buoyant freshwater exerts a stronger influence on the coastal waters east of the estuary, extending to the

vicinity of the 100 m isobath.

The depth-integrated vorticity equations in the cross-isobath direction are adapted to describe the mechanisms shaping the circulation structure in the NSCS. Building on previous findings that the along-isobath pressure gradient force, composed of JEBAR and $PGF_{y*}^b$, represents the dominant driver of the NSCS shelf circulation, this study further clarifies that the baroclinic and river plume anomalies

shape the spatial distribution of cross-isobath velocity anomalies in the coastal waters, while JEBAR primarily governs the region extending from the 100 m isobath to the open ocean. Through further analysis of JEBAR structure, it is found that the bottom density anomaly contributes to positive cross-isobath velocity anomalies, while the density stratification anomaly generates negative values during El Niño years, which collectively influence the characteristic meandering pattern of the

southwestward shelf current. Figure 10 delineates the regions of dominance exerted by the primary drivers: runoff and ENSO ($PGF_{y*}^b$ and JEBAR) during summer in the NSCS, and the main influencing factors are basically arranged from the shore to the open ocean as runoff, $PGF_{y*}^b$, and JEBAR.

Although this study categorizes years based on ENSO phases without considering

phase-dependent dynamics (e.g., developing vs. decaying events), several important aspects remain for future research. First, the phase-locking effects of ENSO could alter seasonal circulation patterns. Second, inter-basin interactions—particularly with the Indian Ocean Dipole (IOD)—deserve further exploration. A positive IOD (characterized by a cool western and warm eastern Indian Ocean) frequently coincides with El Niño, potentially amplifying atmospheric teleconnections that modulate SCS wind patterns and Kuroshio intrusion. Conversely, La Niña conditions, which often occur in conjunction with a negative IOD, may enhance southwesterly monsoon winds, strengthening coastal upwelling and altering freshwater plume dispersal patterns.

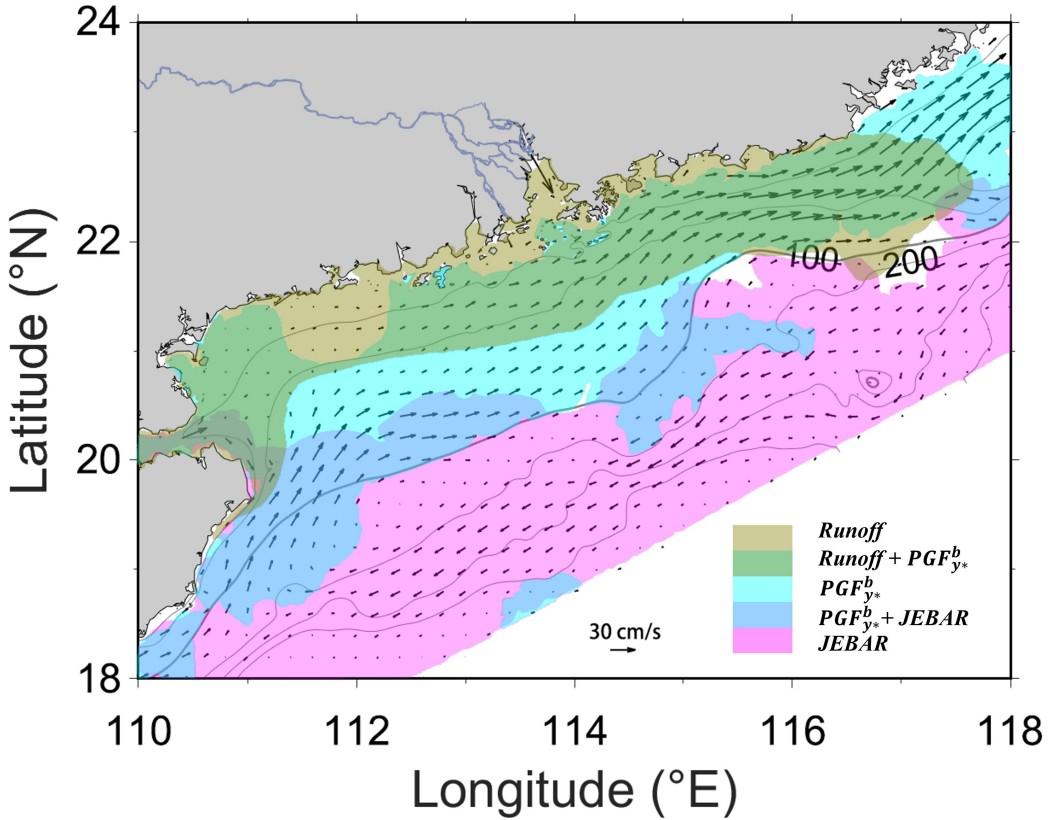

**Figure 10.** Schematic map of the regions controlled by different impact factors (*Runoff*, $PGF_{y*}^b$ and *JEBAR*) during summer in the NSCS. The arrows depict the mean summer shelf circulation.

**Appendix A: Supplementary validation**

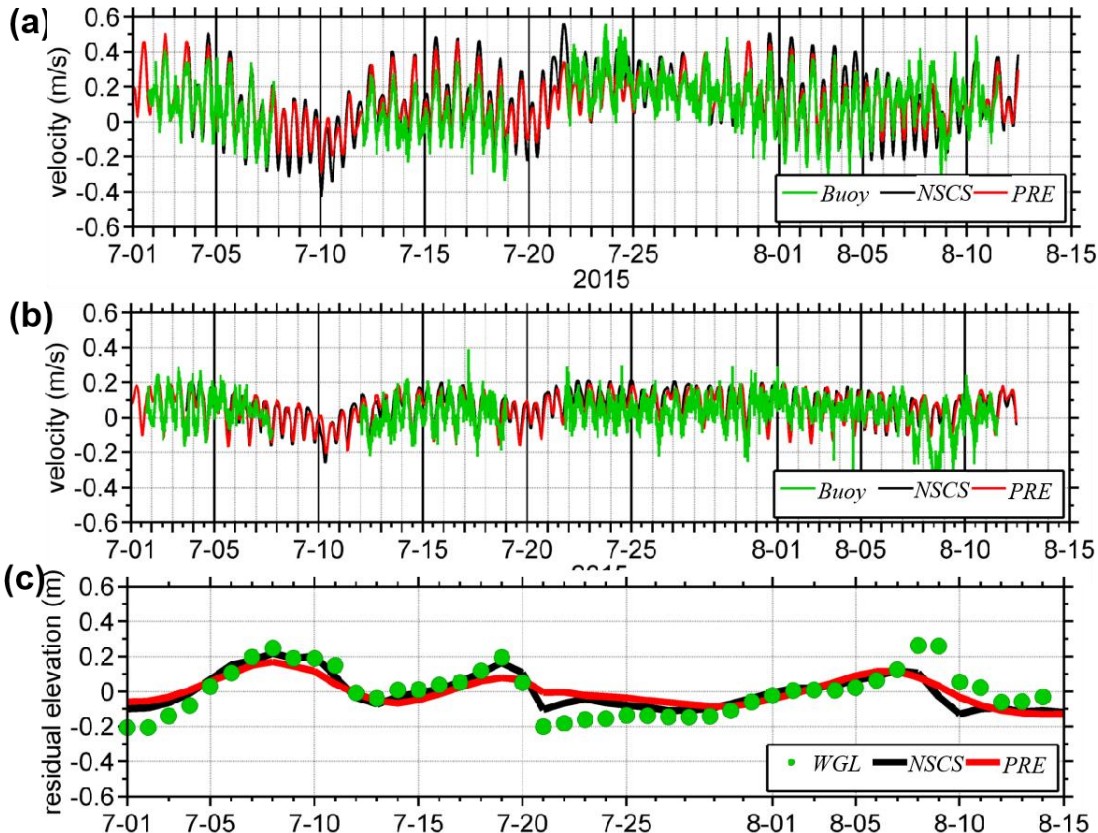

**Figure A1.** Time series of depth-averaged velocities—(a) zonal and (b) meridional—at the buoy/ADCP station, and (c) residual (de-tided) sea level at Waglan Island (WGL) during July–August 2015. Observations are shown in green, the NSCS simulation (this study) in black, and the PRE simulation from Liu et al. (2020) in red. Units: m s⁻¹ for (a–b) and m for (c).

*Data availability.*

The timeseries of long‑term hydrographic data observed in the surface and bottom layers of Hong Kong waters is described in Deng et al. (2022) and is publicly accessible at https://cd.epic.epd.gov.hk/EPICRIVER/marine/?lang=en. The atmospheric forcing data utilized in this study can be obtained from the ECMWF ERA5 dataset (Hersbach et al., 2023), available at https://doi.org/10.24381/cds.f17050d7. The sea surface temperature data are described in Maturi et al. (2017) and openly accessible through NOAA (2024) at https://coralreefwatch.noaa.gov/product/5km/index_5km_sst.php. Gridded sea level anomaly (SLA) and geostrophic velocity data are openly accessible through the EU Copernicus Marine Service (CMEMS), and the relevant DOI link is https://doi.org/10.48670/moi-00148.

*Author contributions*

SYP: Conceptualization, methodology, formal analysis, investigation, writing – original draft, visualization.

LYX: Conceptualization, methodology, resources, data curation, writing – review and editing.

ZP: Software, validation, data analysis, writing – review and editing.

LZQ: Supervision, funding acquisition, project administration, writing – review and editing.

CZY: Investigation, methodology, writing – review and editing, project administration.

All authors have read and approved the final manuscript.

*Competing interests*

The contact authors have declared that none of the authors has any competing interests.

*Acknowledgement*

This project was supported by National Natural Science Foundation of China (42450181, 42276004, 42376024), Science and Technology Development Fund, Macau SAR (File/Project no. 001/2024/SKL,
0040/2023/R1A1, SP2025-00005-CRO), Science, Technology and Innovation Commission of Shenzhen Municipality (JCYJ20240813094213018). The work described in this paper was substantially supported by a grant from the Research Grants Council of the Hong Kong Special Administrative Region, China (AoE/P-601/23-N). CORE is a joint research centre for ocean research between Laoshan Laboratory and HKUST. This work was performed in part at the SICC, which is
supported by the SKL-IOTSC, University of Macau.

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
