# Peer review of "Interannual variability of summertime cross-isobath exchanges in the northern South China Sea: ENSO and riverine influences"

_EGUsphere, 2025_

## Author Comment (AC1)

**Comments from Reviewer 1:**

This manuscript presents a comprehensive investigation of the interannual variability of summer shelf circulation in the Northern South China Sea (NSCS) from 2000 to 2022 based on ROMS modeling. The study makes significant contributions to understanding the differential impacts of ENSO and Pearl River Estuary (PRE) freshwater runoff on NSCS circulation dynamics. While the paper is generally well-structured and scientifically sound, several aspects require clarification and improvement before publication.

**Response:** We sincerely thank Reviewer for the positive assessment and the thoughtful suggestions. In response, we have revised the manuscript to improve clarity and robustness. Specifically, we clarified the methodological choices and diagnostics, strengthened the statistical treatment and significance testing, refined attributions to ENSO and PRE runoff, and improved figure readability and terminology consistency. All changes are tracked in the revised file; below, we respond point-by-point and indicate where each revision appears.

1.  Line 57-66, I recommend incorporating this study's findings and conclusions to more explicitly identify the shortcomings of existing research in quantifying the impacts of ENSO and river runoff on cross-shelf transport in the northern South China Sea.

**Response:** Thank you for this helpful suggestion. We agree that the Introduction should more explicitly articulate where prior studies fall short in quantifying the respective impacts of ENSO and Pearl River runoff on cross-shelf transport in the NSCS. We have revised the last paragraph of the Introduction (Lines 57 – 66) to (i) summarize specific gaps in the literature and (ii) state how our analysis addresses those gaps. The new text is quoted below.

[Line 59-82]: Previous studies of NSCS shelf circulation have primarily examined

seasonal patterns and their wind-driven dynamics (Hu, 2000). Short-term summer variability in the shelf current has also been explored: Geng et al. (2024) demonstrated that tides modulate sea surface height, influencing shelf pressure gradients, while Liu et al. (2020) showed that cross-isobath exchanges during upwelling and downwelling winds are sensitive to along-shelf pressure gradients shaped by complex bathymetry. The intensity of this upwelling exhibits notable interannual variability, often linked to the El Niño‐Southern Oscillation (ENSO) (Shu et al., 2018). For example, during the summer of 1998—an El Niño year—enhanced alongshore wind stress substantially intensified coastal upwelling (Jing et al., 2011). Despite these advances, the interannual variability of summer shelf circulation remains poorly constrained, particularly in quantifying the depth structure of cross-isobath transport and attributing it to distinct forcings. While expanded observational datasets, satellite products, and high-resolution modeling (Hong & Wang, 2008; Shu et al., 2011; Zu et al., 2020) have refined our understanding of NSCS dynamics, the combined effects of ENSO, PRE runoff, and regional current meandering have rarely been assessed within a unified, shelf-wide framework that computes cross-isobath transport and partitions the pressure-gradient forcing (e.g., JEBAR vs. bottom-related terms).

This study investigates the interannual variability of NSCS shelf circulation during summer from 2000 to 2022, using long-term observations and numerical modeling. We examine how ENSO modulates regional atmospheric and oceanic forcing, how the PRE runoff controls freshwater plume behavior, and how vorticity-related processes—including JEBAR, bottom stress curl, and nonlinear vorticity advection—govern cross-isobath exchanges and meandering shelf currents. Our study also provides a quantitative attribution of ENSO and river runoff impacts on cross-shelf transport in the NSCS, by explicitly computing depth-resolved cross-isobath transport (positive onshore) and decomposing the pressure-gradient/vorticity terms (JEBAR, bottom stress curl, nonlinear relative vorticity advection). By integrating these processes, our study provides new insight into how regional and remote forcings

interact to shape NSCS shelf circulation, with implications for hydrographic structure and ecosystem variability.

2. Section 2, the authors have presented comparison results between observations and model simulations for temperature, salinity, and sea level. However, they did not mention the more crucial current observation data. How does the model perform in simulating shelf circulation? Of course, I understand that current observation data is relatively scarce, and model simulations of circulation may have greater errors. It would be preferable to provide some validation of circulation observations here.

**Response:** Thank you for underscoring the importance of current validation. We agree and have added a concise comparison between our NSCS simulation and summertime high-resolution buoy/ADCP observations over the shelf off the Pearl River Estuary reported by Liu et al. (2020). Given the scarcity of public inner-shelf current records, we re-use that dataset and evaluate our simulation for the same location and period. As shown in Fig. R1a – b, the depth-averaged along- and cross-shelf (zonal/meridional) velocities reproduce the observed velocities fluctuations with realistic amplitude and phase. The residual (de-tided) sea level at Waglan Island likewise exhibits consistent subtidal variability between observations and the model (Fig. R1c). Together with the SST/SSS/SLA validations in Section 2, these checks indicate that the model represents the summertime shelf circulation with sufficient skill for the diagnostics presented here, while we acknowledge greater uncertainty on the innermost shelf due to limited observations.

We introduce this supplementary plot in the revised Section 2:

Moreover, to assess currents where observations exist, we compare our simulation with summertime high-resolution ADCP records over the shelf off the Pearl River Estuary reported by Liu et al. (2020) for the same site and period, and we examine residual (de-tided) sea level at the Waglan Island tide gauge. The simulated along- and

cross-shelf velocities reproduce the observed fluctuations with realistic amplitude and phase, and the residual sea level shows consistent subtidal variability (Fig. A1).

[Figure]

Figure A1. Time series of depth-averaged velocities—(a) zonal and (b) meridional—at the buoy/ADCP station, and (c) residual (de-tided) sea level at Waglan Island (WGL) during July–August 2015. Observations are shown in green, the NSCS simulation (**this study**) in black, and the PRE simulation from Liu et al. (2020) in red. Units: m s⁻¹ for (a–b) and m for (c).

**Reference:**

Liu, Z. and Gan, J.: A modeling study of estuarine-shelf circulation using a composite tidal and subtidal open boundary condition, Ocean Modelling, 147, doi:10.1016/j.ocemod.2019.101563, 2020.

3.  Section 3.1, the MVEOF analysis serves as a critically important tool in this study. However, for readers unfamiliar with this methodology, the current paper provides insufficient explanation. I had to search online to understand the fundamental principles of this method before being able to properly follow the article's discussion. The introduction of MVEOF should be moved to Section 2 (Methods), and requires more detailed explanation.

**Response:** Thank you for this constructive suggestion. We agree that a brief methodological explanation of MVEOF will help readers unfamiliar with the approach. In the revision, we have moved the MVEOF description to Section 2 (Methods) and added the following concise paragraph. Section 3.1 now simply applies the method and refers back to Methods. In the revised section 3.1, we included: We use Multivariate Empirical Orthogonal Function (MVEOF) to extract the dominant coupled spatio-temporal modes shared by multiple, related variables. The method extends conventional EOF by forming a joint covariance structure that includes cross-covariances among the selected fields (e.g., sea-surface height, temperature, and velocity), thereby identifying patterns that maximize joint variance across variables. Each mode is paired with a principal-component (MVPC) time series that describes its temporal evolution, and with variable-wise spatial maps (from regressing the MVPC back onto each field) that show how the variables co-vary in space.

4.  Line 147, there is an extra closing parenthesis ")" in this line.

**Response:** Thanks for your scrutiny, and it is corrected in this revised version.

5.  All variables in the equations should be clearly defined in the text. For example, the H and tao in Equ. 2.

**Response:** We are sorry for this inadvertent omission, and the definitions of the items

have now been fully restored in the revised manuscript.

6. As a researcher specializing in shelf material transport, I find the subject of this paper particularly compelling. The study provides in-depth dynamic analysis of ENSO and runoff impacts on cross-isobath transport, but lacks quantitative information that would be most useful for practical applications. For instance: (1) Missing baseline metrics: What is the approximate summer cross-shelf transport velocity in the northern South China Sea? This could be calculated from Figure 4b. Without concrete values, the results are difficult for other researchers to directly utilize. (2) Quantification of forcing contributions: Can the relative contributions of ENSO versus river discharge to cross-isobath transport be quantified? Including these quantitative results in the abstract or conclusions would significantly enhance the paper's citation potential and appeal to a broader scientific audience.

**Response:** Thank you for emphasizing the need for practical, quantitative guidance. We share the concern that a single "baseline number" can be misleading for a heterogeneous shelf system: cross-isobath velocity varies with depth, local bathymetry, wind/tide events, and model configuration (e.g., resolution). To balance interpretability and uncertainty, we now report order-of-magnitude ranges using $O(\cdot)$ notation rather than decisive values. Specifically, summertime cross-isobath flow is $O(10^{-2}\text{ m/s})$ on average, with localized peaks up to $O(10^{-1}\text{ m/s})$ near dynamical hotspots (e.g., Taiwan Shoal; Fig. 4b). This conveys practical magnitude while avoiding over-precision.

Regarding the relative influences of ENSO and Pearl River discharge, we now emphasize a domain-aware qualitative partition — PRE discharge as the principal control on the inner/mid shelf and ENSO exerting a larger influence toward the slope — supported by the MVEOF co-variability patterns and the dynamical decomposition. Because percentage splits are sensitive to episodic events, we refrain from quoting a

single percentage without a full uncertainty framework; instead, we highlight where each forcing is most influential and note that the sign and spatial patterns are robust across years.

---

## Author Comment (AC2)

**Comments from Reviewer 2:**

This manuscript studies the interannual variability of shelf circulation in the Northern South China Sea (NSCS). Low-frequency variability in the ENSO and the Pearl River Estuary (PRE) runoff are shown to have important effects in the spatial patterns of cross-isobath transport. Distinct regimes exist inshore and offshore of the 100 m isobath, respectively associated with bottom friction+nonlinear effects and stratification-dominated effects. The regions east and west of the PRE's outflow are also starkly different, being respectively dominated by variability in the PRE plume's volume and in the Kuroshio's intrusions.

The text, figures and tables could use minor improvements but read generally well, and the reasoning is easy to follow. The major issues I see with the manuscript are in terms of a couple of subjective choices, namely the regression analysis methodology and the criterion for identifying large-outflow years.

**Response:** We appreciate the reviewer's thoughtful comments on our work. We have taken your concerns into account and have made the appropriate revision following your suggestions.

Major points:

M1 (lines 218-229): I am not sure I follow the need for the regression step in the two-stage regression approach. My understanding is that this analysis is a conditional average of the anomaly fields for each variable at the times when each variable's MVPC1/PC1 was in a positive phase, is that correct? I do not follow where the linear slopes calculated from the least-squares analysis are actually used. The time series of the MVPC1/PC1 should contain the relevant temporal variability of the leading EOF mode. Please clarify this paragraph.

To justify the need for a more elaborate method, the authors also need to compare it to the simplest one. How do all results in the paper compare to doing the same analyses

just by conditionally-averaging the anomaly fields over years of positive MVPC1/PC1 phase?

**Response:** We appreciate the request to clarify the role of the regression step. Our aim is to (i) extract the spatial pattern linked to the leading coupled mode and (ii) present a representative positive-phase amplitude with reduced year-to-year noise. Concretely, we regress each anomaly field on the standardized MVPC1/PC1; the resulting slope map is an effect size per +1 s.d. of the mode. For display, we scale that slope by the mean positive-phase amplitude to form the plotted "positive-phase" pattern. Thus, the regression slopes are directly used to construct the maps and quantify amplitudes.

Following the reviewer's suggestion, we also computed simple conditional averages of the raw anomalies over positive-phase summers. These maps closely resemble the regression-scaled patterns, with only small coastal differences. We now note this agreement in the text and provide a short side-by-side panel in the Supplement.

In the revised manuscript, we include these information as:

To characterize interannual anomalies in shelf currents and hydrography, we use a regression-scaled composite. First, for each variable, we regress its anomaly field on the standardized MVPC1/PC1 (Fig. 3g,h). The resulting slope map is an effect size－anomaly per +1 standard deviation of the mode－that uses all years and reduces synoptic noise. To depict the positive phase, we scale this slope by the mean of the index during summers when the index is positive (years in Tables 1－2), which is numerically close to a simple conditional average but provides a directly interpretable amplitude and enforces phase antisymmetry. For transparency, we also compute simple conditional-average maps; these closely match the regression-scaled patterns (Fig. S2), and we therefore retain the regression-scaled maps in the main text for consistency across variables.

[Figure]

**Figure S2.** (a) Mean velocity vector anomalies (cm s⁻¹), and (b) mean cross-isobath velocity anomalies (cm s⁻¹) during summer in positive MVPC1 years (Table 1).

M2 (line 197-199): Is there an objective criterion for choosing large-runoff years, like an outflow volume threshold? I think it is important to have one, and it should be described here.

Because the choice of the threshold is also arbitrary (e.g., it could be the years where the outflow was greater than the 75th or 90th percentile), a second step is to study the sensitivity of the results to this choice as well.

**Response:** We appreciate this suggestion. We now define large-runoff years using a simple percentile rule. For each year $y$, we compute the summer-mean Pearl River discharge $Q(y)$. Years with $Q(y) \geqslant Q_p$ are classified as high-runoff, where we use $p=70\%$ as a reference choice. In our record (2000–2022) this corresponds to 26,000 m³ s⁻¹, yielding **7** high-runoff and **6** low-runoff summers. To assess sensitivity, we repeat the key composites and regressions for $p=80\%$. The principal spatial features and interpretations are generally similar across thresholds, with some variations in amplitude as expected. We also compare this percentile rule with the sign of the SSS-EOF PC1 and find substantial overlap. These details are described in Methods and summarized in Fig. S1.

In the revised Supplement, we included:

The large-runoff years are identified by a simple percentile rule: for each year we

compute the summer-mean Pearl River discharge $Q(y)$, and label years with $Q(y) \geq Q_{70\%}$ as high-runoff. Over 2000–2022, this threshold equals 26,000 m³ s⁻¹. Figure S1 shows that raising the cutoff to $Q_{80\%}$ produces almost the same spatial patterns with only minor amplitude differences, which confirms that $Q_{70\%}$ is sufficient to distinguish between large- and low-runoff years.

[Figure]

**Figure S1**. (a) Salinity anomaly profiles (psu) during summer in years exceeding the 80% runoff threshold (Table 2) at Transect A; (b) salinity anomaly profiles during summer in years exceeding the 80% runoff threshold at Transect B; (c) salinity anomaly profiles during summer in years exceeding the 80% runoff threshold at Transect C.

M3: I think a key result worth emphasizing is the identification of the different dynamical regimes in terms of their response to different ENSO/PRE plume drivers (inshore PGF_{y*}^b-dominated/offshore JEBAR-dominated and west Kuroshio intrusion-dominated/east PRE plume-dominated). I think adding a schematic/cartoon-type figure illustrating these would be a good way to summarize the results in a mechanistic way and make them more visible to readers studying other regions influenced by Western Boundary Currents, wind-driven upwelling, and large river outflows.

**Response:** We appreciate the suggestion. We have added a concise schematic that summarizes the summer dynamical regimes identified in our analyses (new Fig. 10). The schematic highlights (i) an inshore branch where bottom pressure-gradient forcing dominates cross-isobath transport, (ii) an outer-shelf/slope branch where JEBAR is predominant, and (iii) a west–east contrast between a

Kuroshio-intrusion-influenced western sector and a PRE-plume-influenced eastern sector. We also indicate the sign of the anomalies associated with ENSO phase and with high/low discharge in a qualitative way to avoid visual clutter. This figure is intended as a compact, mechanistic summary for readers and complements the quantitative maps and budgets in the main text.

In the revised manuscript, we included:

Figure 10 delineates the regions of dominance exerted by the primary drivers: runoff and ENSO ( $PGF_{y*}^{b}$ and JEBAR) during summer in the NSCS, and the main influencing factors are basically arranged from the shore to the open ocean as runoff, $PGF_{y*}^{b}$, and JEBAR.

[Figure]

**Figure 10.** Schematic map of the regions controlled by different impact factors (*Runoff*, $PGF_{y*}^{b}$ and *JEBAR*) during summer in the NSCS. The arrows depict the mean summer shelf circulation.

Minor points

m1 (lines 39-41): Topographic effects should be more important in locations with more curved isobaths such as in the NSCS' widened shelf area, as previous work in the NSCS shows. So I don't follow why nearly shore-parallel isobaths should result in enhanced cross-isobath flow.

**Response:** We apologize for any confusion caused by the ambiguous expression. This sentence has been rewritten for clarity in the revised version. The last sentence of the introduction section now reads:

" In addition to wind-driven Ekman transport, cross-isobath exchanges are strongly influenced by topographic effects, which mainly function over the concave shelf where the isobaths show spatial irregularity (Liu et al., 2020)."

m2 (lines 82 and 119): Comparing the model resolution to the local first deformation radius derived from the model stratification is important here, especially inshore of the 100 m isobath (where the nonlinear terms are shown to be more important in the depth-averaged vorticity balance).

**Response:** The local first deformation radius inshore of the 100 m isobath (apart from the coastal area) ranges from a few to ~10 km, which is sufficiently larger than the model resolution yet at least an order of magnitude smaller than the shelf width, so the nonlinear terms likely play only a minor role in the depth-averaged vorticity balance.

In the revised manuscript, we included at end of section 3.2:

It is also noteworthy from this figure that landward of the 100-m isobath (excluding the immediate nearshore), the first baroclinic Rossby radius exceeds the model grid spacing yet remains an order of magnitude smaller than the shelf width, supporting the validity of the climatic scales adopted in this study.

m3 (line 87): A couple of example references using the Mellor-Yamada scheme could be added here, in addition to the original paper describing the scheme.

**Response:** Two representative papers (Gan et al., 2006; Jing et al., 2009) employing the Mellor–Yamada scheme for South China Sea circulation modeling have now been cited in the revised manuscript.

m4 (Fig 1): It would be helpful to add the 100 m and 200 m isobaths to this figure for reference. In Figs. 4, 6, and A1, it would also help to have them labelled on the figure itself.

**Response:** Thanks for this important reminder. We have added the isobaths and labels in the figures of the revised manuscript.

Such as Figure 1:

[Figure]

m5 (Fig 1's caption): Are the geostrophic currents shown as pink arrows the surface

geostrophic velocity derived from the model sea surface slopes? This could use clarification. Differentiating between "geostrophic" and "shelf" currents is also confusing because there are geostrophic currents both on the shelf and offshore.

**Response:** The geostrophic currents outside the model domain are derived from CMEMS satellite products. A clearer statement has been added to the revised manuscript. The caption of Fig. 1 now reads:

"Figure 1. Overview map of the northern South China Sea. Model-simulated shelf currents (colored arrows) within the model domain and satellite-derived geostrophic currents (pink arrows) outside the scope of the model simulation represent summer averages from 2000 to 2022. The upper panel shows the corresponding climatological wind field. Red markers indicate the observational sites used for model validation in Fig. 2. The upper left sub-graph shows the location of the northern South China Sea."

m6 (line 247): How are the degrees of freedom estimated (e.g., from integral timescales derived from the time series of the velocity components at each grid point)? It would be good to describe it here.

**Response:** The degrees of freedom estimated are automatically computed by the MATLAB toolbox. For the simple linear regression $Y = \beta_0 + \beta_1 X + \varepsilon$ with N observations, the model estimates two parameters (intercept and slope). Consequently, the residual degrees of freedom are N-2. The caption of Fig. 4 now reads:

"Figure 4. Depth-averaged circulation and associated anomalies in the northern South China Sea: (a) mean velocity vectors (cm s⁻¹) and (b) mean cross-isobath velocity (cm s⁻¹) averaged over summers from 2000 to 2022, where positive cross-isobath velocity indicates flow from deeper to shallower waters; (c) regression map of velocity vector anomalies (cm s⁻¹), and (d) regression map of cross-isobath velocity anomalies (cm s⁻¹) during positive MVPC1 years (Table 1). Shaded areas in (c) and (d) denote regions where the 90% confidence level is not met, based on a two-tailed t-test using

the estimated standard deviation and sample degrees of freedom. For the simple linear regression, the degrees of freedom are N-2 (N is the number of observations). The two-stage regression approach is detailed in Section 3.2. Positive values in (c) and (d) indicate flow toward shallower waters."

m7 (Fig. 5) The discussion relies on different stratification regimes, which appear to be both salinity- and temperature-driven. It would therefore help to overlay isopycnals of the conditionally-averaged density fields on each panel.

**Response:** Thanks for your suggestion. We have now overlaid contour lines of the regressed density anomaly on each panel in the revised Fig.5:

[Figure]

**Figure 5.** Regression maps of hydrographic anomalies along three cross-shelf transects (locations shown in Fig. 4b): (a–c) temperature anomaly profiles (°C) during positive MVPC1 years (Table 1) at Transects A, B, and C, respectively; (d–f) salinity anomaly profiles (psu) during positive PC1 years (Table 2) at the same transects. The isolines depict the regressed density anomalies for the corresponding scenarios. The two-stage regression approach is detailed in Section 3.2.

m8 (line 290-291): It is more objective to include some metric of the smallness of the

GMF term, for example, what is its size relative to the next-largest term in the balance? This ratio will also vary spatially, so I suggest the authors include a figure with the GMF term's spatial structure and its relative size in the Appendix.

**Response:** Thanks for your suggestion. A figure illustrating the contribution of the GMF term (Fig. S3) has been added to the revised manuscript.

In the revised Supplement, we included:

The spatial structure of the cross-isobath velocity anomaly contributed by the term GMF during positive MVPC1 years is displayed in Fig. S3. Its magnitude is negligible compared with the other terms (except within the PRE). Therefore, it is reasonable for us to omit the discussion of this item in the main text.

[Figure]

**Figure S3.** Regression maps of horizontal cross-isobath velocity anomalies (cm s⁻¹) during positive MVPC1 years (Table 1), attributed to the gradient of momentum flux (GMF).

m9 (Fig. 8 caption): Are these each of the JEBAR terms' contributions to the total PGF_{y*}^b/f anomalies in cm/s (like Fig. 6)? Please add the units to the caption like in Fig. 6, or to the colorbar labels.

**Response:** All items in Fig.8 were calculated independently and should be expressed in m² s⁻². This unit has now been included in the figure caption in the revised manuscript. The caption of Fig. 8 now reads:

"**Figure 8.** Regression maps of horizontal components (m² s⁻²) in the JEBAR term during summer and positive MVPC1 years, under different runoff conditions: (a–c) correspond to positive PC1 years, and (d–f) to negative PC1 years. Panels show (a, d) the full baroclinic gradient term ($\frac{g}{\rho_0}\int_{-H}^{0} z(\rho - \bar{\rho})dz$), (b, e) the contribution from vertical density stratification ($-\frac{g}{\rho_0}\int_{-H}^{0} \frac{z^2}{2}\frac{\partial(\rho-\bar{\rho})}{\partial z}dz$), and (c, f) the contribution from bottom density anomaly ($-\frac{g}{\rho_0}\frac{H^2}{2}(\rho - \bar{\rho})_b$). The MVPC1 time series is shown in Fig. 3g, with corresponding positive-phase years listed in Table 1, while the PC1 time series is shown in Fig. 3h with its positive-phase years listed in Table 2. The two-stage regression approach is detailed in Section 3.2."

Typos/minor edits

Line 23: Intrusion -> intrusions

Line 91: Large amount of freshwater influx -> a large freshwater influx

Line 119: Smaller scaled -> smaller-scale

Line 119: Could be further detailed -> are not fully resolved

Line 196: Streamflow -> runoff/outflow

Line 214: Missing space before "Interannual"

Line 294: Within the -> inshore of

Line 377: In the -> in

**Response:** All corrections have been applied in the revised manuscript.

**References:**

Gan J., Li H., Curchitser E. N., and Haidvogel D. B.: Modeling South China Sea circulation: Response to seasonal forcing regimes, Journal of Geophysical Research: Oceans, 111(C6). doi:10.1029/2005JC003298, 2006.

Jing Z., Qi Y., Hua Z., and Zhang H.: Numerical study on the summer upwelling system in the northern continental shelf of the South China Sea, Continental Shelf Research, 29(2): 467–478, doi:10.1016/j.csr.2008.11.008, 2009.

---

## Author Comment (AC3)

Dear Authors

Thank-you for your revised manuscript. In view of the substance of some of their comments and the considerable ensuing changes, I intend to send your revised manuscript back to the reviewers in due course. However, I think there are some points that it is best to attend to first. Please see "Detailed comments" below. Yours sincerely

John Huthnance (editor)

**Response**: Thank you for your careful handling of our revised manuscript. We carefully responded to the detailed comments and revised the manuscript accordingly.

**Detailed comments**

1. Lines 138-144 and 157. I think a reference specific to MVEOF would help. Your inserted text in section 2 states very little extra to what was previously in section 3.1 (c.f. Referee 1 comment)

   **Response**: Thanks for the suggestion from reviewer. Following your suggestion, we give reference for MVEOF.

   We use Multivariate Empirical Orthogonal Function (MVEOF) to extract the dominant coupled spatio-temporal modes shared by multiple, related variables (Dawson, 2016; Liang et al., 2018)

   Reference:

   Dawson, A.: eofs: A library for EOF analysis of meteorological, oceanographic, and climate data, J. Open Res. Softw., 4, 1, https://doi.org/10.5334/jors.122, 2016.

   Liang, Y.-C., Mazloff, M. R., Rosso, I., Fang, S.-W., and Yu, J.-Y.: A multivariate empirical orthogonal function method to construct nitrate maps in the Southern Ocean, J. Atmos. Ocean. Tech., 35, 1505–1519, https://doi.org/10.1175/JTECH-D-17-0201.1, 2018.

2. Lines 143-144. If the maps are for individual variables, is the covariation in space only subjective by looking at two of the individual variable maps.

   **Response**: The individual maps are presented for each variable; however, they are all derived from the same MVPC time series (i.e., the same mode). Each map reflects how a given variable spatially expresses the same underlying temporal pattern. As a result, the spatial patterns across variables represent a consistent, objectively derived co-varying structure—rather than a purely subjective visual comparison.

   In the revised manuscript, we have clarified this point as follows: "These maps are derived from the same MVPC time series. The spatial patterns across variables reflect a consistent, objectively derived co-varying structure."

3. Lines 158-159 and Figure 4 caption. Referee 2 comment m6 is implicitly pointing out that the number N of observations is not the correct starting point for determining degrees of freedom. If successive observations are correlated, because the time scale is longer than the sampling interval, then there are fewer degrees of freedom.

   **Response**: Thanks for further clarification of this issue. We revised our response to Referee 2 and the manuscript accordingly.

   Updated response to Referee 2:

   "The degrees of freedom estimated are automatically computed by the MATLAB toolbox. For the simple linear regression $Y=\beta_0+\beta_1 X+\varepsilon$ with N observations, the model estimates two parameters (intercept and slope). Consequently, the residual degrees of freedom are N-2.

   This calculation assumes independent temporal observations. Here we acknowledge that the temporal autocorrelation in the time series may reduce the effective degrees of freedom, thus affect the confidence level. In this study, our emphasis is on the broad spatial patterns of the regressed velocity vector anomalies and cross-isobath velocity anomalies, rather than marginal differences

at the confidence threshold. Nonetheless, we have added a note in the manuscript to acknowledge this uncertainty.

The caption of Fig. 4 now reads:

"Figure 4. Depth-averaged circulation and associated anomalies in the northern South China Sea: (a) mean velocity vectors (cm s⁻¹) and (b) mean cross-isobath velocity (cm s⁻¹) averaged over summers from 2000 to 2022, where positive cross-isobath velocity indicates flow from deeper to shallower waters; (c) regression map of velocity vector anomalies (cm s⁻¹), and (d) regression map of cross-isobath velocity anomalies (cm s⁻¹) during positive MVPC1 years (Table 1). Shaded areas in (c) and (d) denote regions where the 90% confidence level is not met, based on a two-tailed t-test using the estimated standard deviation and sample degrees of freedom. For the simple linear regression, the degrees of freedom are N-2 (N is the number of observations), assuming independent observations. As temporal autocorrelation may reduce the effective number of independent samples, the reported regions with 90% confidence level may include uncertainties. The two-stage regression approach is detailed in Section 3.2. Positive values in (c) and (d) indicate flow toward shallower waters."
"

4. Lines 205, 321. "inside/within the 100 m isobath" –> "inshore of the 100 m isobath" (c.f. Ref.2)

   **Response**: Corrected and thanks!

5. Line 219. "(70% runoff threshold of these years)" –> "(70th percentile for these years)"? and Lines 219-220 "while a sensitivity test of the selected thresholds is shown" –> "while sensitivity to the choice of percentile is shown"?

   **Response**: Corrected and thanks!

6. Equation (2) and Line 314. Presumably $\xi$ is relative vorticity but I miss explicit definition.

   **Response**: Thanks for the reminder, the $\xi$ is relative vorticity and we added the definition in the revised manuscript

   "$\psi$ is the transport streamfunction, $\xi$ is the relative vorticity, and $\vec{v}(\bar{u}, \bar{v})$ denotes the depth-averaged velocity, respectively"

7. Lines 317-318. In response to Referee 2 comment m8 I think you need to refer to the Supplement and figure S3 here, otherwise the reader does not see why GMF is excluded.

   **Response**: Following reviewer's suggestion, we clarify this in the revised manuscript

   "The cross-isobath velocity anomaly contributed by the term GMF during positive MVPC1 years is negligible compared with the other terms (Fig. S3), thus it is excluded from the following discussion."

8. Line 390. Better to omit "is strong or weak"

   **Response**: Thanks for the suggestion and we omit it from the revised manuscript

9. Line 414. "is the exceptionally high core of density anomaly" –> "is the core of exceptionally high density anomaly"?

   **Response**: Corrected and thanks!

---

## Author Response (AR3)

Dear Authors.

I now have referees' comments on your revised manuscript. One is content, the other asks for minor revision; their comments are copied below with some <> of mine. Please respond to all these and upload a re-revised manuscript.

Yours sincerely

John Huthnance (editor).

**Response:** We sincerely thank you for your continued and careful handling of our manuscript, as well as for your constructive suggestions throughout the review process. In this round, we have carefully addressed all comments from the referee and revised the manuscript accordingly. A re-revised version, together with a detailed point-by-point response to the reviewer's comments and your notes in angle brackets, has been uploaded for your consideration.

Referee comments with some <>

I thank the authors for their thorough work addressing my comments on the manuscript's previous version. I believe the revised version has certainly improved, but I still have some points to raise.

**Response:** We sincerely thank the reviewer for the careful re-evaluation of our manuscript and for the constructive comments. We appreciate your recognition of the improvements in the revised version and have carefully considered all remaining concerns. Corresponding revisions have been made in the manuscript, as detailed in our point-by-point responses below.

Authors' response to point m2: "The local first deformation radius inshore of the 100 m isobath (apart from the coastal area) ranges from a few to ~10 km, which is sufficiently larger than the model resolution yet at least an order of magnitude smaller than the shelf width, so the nonlinear terms likely play only a minor role in the depth-averaged vorticity balance. In the revised manuscript, we included at end of section 3.2: It is also noteworthy from this figure that landward of the 100-m isobath (excluding the immediate nearshore), the first baroclinic Rossby radius exceeds the model grid spacing yet remains an order of magnitude smaller than the shelf width, supporting the validity of the climatic scales adopted in this study."

In Fig. 6b,c, it is shown that the Relative Vorticity Advection (RVA) nonlinear term dominates the bottom along-isobath pressure gradient force's ($PGF_{y*}^b$) structure. Comparing Fig. 6b and Fig. 6c shows that the $PGF_{y*}^b$ and the RVA actually have similar magnitudes and

spatial patterns just inshore of the 100 m isobath, as stated by the authors in lines 325-329 of the revised manuscript. This seems to contradict the authors' response to this point, as the nonlinear term does play a major role in the depth-averaged vorticity balance in those areas.

**Response:** We apologize for the confusion caused by our earlier wording.

Our statement that the nonlinear terms "likely play only a minor role" was intended to refer to a scale analysis of the depth-averaged momentum equations, in which the Rossby number over the NSCS shelf is small and the intensity of advective nonlinearity is weaker than intensity of the Coriolis and pressure-gradient terms, particularly in the cross-shelf momentum balance. In this sense, the large-scale shelf circulation remains close to geostrophic, and the nonlinear terms are secondary in setting the overall momentum balance.

By contrast, Fig. 6b–c shows the decomposition of the bottom along-isobath pressure-gradient term in the depth-averaged vorticity equation. In this vorticity budget, the Relative Vorticity Advection (RVA) term (in the right-hand-side of the $PGF_{y*}^b$ equation) is indeed comparable to, and locally dominates, the structure of $PGF_{y*}^b$ just inshore of the 100 m isobath, which explains the similar magnitudes and spatial patterns of RVA and $PGF_{y*}^b$ in that region. This analysis is used to diagnose how nonlinear advection helps maintain the along-isobath pressure-gradient structure, rather than to suggest that nonlinear terms dominate the total pressure-gradient forcing or overturn the near-geostrophic large-scale balance.

To avoid ambiguity, we have revised the relevant sentence in Sections 3.2 and 4 to clarify that nonlinear terms are dynamically secondary in the large-scale momentum balance, but can still be the leading contributor to the $PGF_{y*}^b$ and thus to the detailed structure of cross-isobath velocity anomalies. We believe this clarification reconciles our previous statement with the patterns shown in Fig. 6.

Line 300-303: "At the climatic scales in this study, the Rossby number is small, so nonlinear advection remains dynamically secondary in the depth-averaged momentum balance, and the large-scale shelf circulation is close to geostrophic, particularly in the cross-shelf direction."

Line 338-340: "Thus, while nonlinear advection is secondary in the large-scale depth-averaged momentum balance, it can still emerge as the leading contributor to the $PGF_{y*}^b$ and thereby shape the detailed structure of cross-isobath velocity anomalies over the coastal band."

A shelf deformation radius of a few kilometers up to ~10 km sounds like the scale one would expect, but the authors need to specify where this estimate is derived from and how it was calculated (e.g., by solving an eigenvalue problem based on model vertical density profiles).
<>

**Response:** We thank the reviewer for raising this important point and for prompting us to clarify the derivation of the first baroclinic deformation radius. In this study, we estimated the baroclinic Rossby radius using the standard two-layer approximation: $R_0 = \frac{\sqrt{g'H}}{f}$. Here, $H$ is the water depth, $f$ represents the Coriolis parameter, and the reduced gravity $g' = \frac{\Delta\rho}{\rho}$ ($\Delta\rho$ could be approximated as the density difference between the upper and lower layers, and $\rho$ denotes the domain-averaged density as reference density). Using this formulation, the deformation radius over most of the shelf indeed falls within a few kilometers to ~10 km.

It is now included in line 296-299: ($\sqrt{g'H}/f$, where $H$ is the water depth, $f$ represents the Coriolis parameter, and reduced gravity: $g' = \Delta\rho/\rho$. $\Delta\rho$ is the density difference between the upper and lower uniform layers, and $\rho$ denotes the domain-averaged density as reference density) ranging from a few to ~10 km,

Point m6 (statistical significance of the regression maps): Considering that most of the manuscript's results (Figs. 4-9) are based on regression maps of various flow diagnostics, I still think it is important to accurately determine the areas where the regression maps are significant (i.e., the gray area indicated in Fig. 4c,d [and Fig. 6c,d?]) by accounting for temporal correlation.

**Response:** We thank the reviewer for highlighting the importance of properly assessing statistical significance in the regression maps. In our analysis, each data point corresponds to a summer-mean (June–August) value, so the regression is performed on interannual seasonal means rather than on monthly or daily time series. Our primary working assumption is therefore that successive summers can be treated as approximately independent realizations for the purpose of a first-order linear regression and *t-test*.

Following this assumption, we used a standard two-tailed *t-test* with N-2 degrees of freedom, which is explicitly stated in the caption of Fig. 4. At the same time, we fully recognize that temporal correlation at interannual scales may reduce the effective number of independent samples. To reflect this, the manuscript notes that: "For the simple linear regression, the degrees of freedom are N − 2 (N is the number of observations), assuming independent observations. As temporal autocorrelation may reduce the effective number of independent samples, the reported regions with 90% confidence level may include uncertainties." (Figure Captions of Fig. 4 and 6-9).

We have chosen a relatively modest confidence level (90%) and use the significance mask primarily to de-emphasize noisy areas (e.g., parts of the continental slope), rather than to draw strong conclusions from marginal features. The large-scale patterns emphasized in the text (such as the meandering shelf current and the JEBAR-dominated offshore response) are robust to reasonable changes in the significance threshold and to the precise treatment of temporal autocorrelation.

Finally, our approach is consistent with previous shelf-circulation studies that applied linear regression to interannual or seasonal-mean quantities without explicitly correcting for temporal autocorrelation in the *t-test* (e.g., Rosentraub & Brenner, 2007; Lentz, 2022). We have clarified this rationale and our assumptions in the revised manuscript to make the limitations and interpretation of the significance maps transparent to readers.

**References:**

Rosentraub Z, Brenner S. Circulation over the southeastern continental shelf and slope of the Mediterranean Sea: direct current measurements, winds, and numerical model simulations. Journal of Geophysical Research: Oceans. 2007 Nov;112(C11).

Lentz SJ. Interannual and seasonal along-shelf current variability and dynamics: seventeen years of observations from the southern New England inner shelf. Journal of Physical Oceanography. 2022 Dec;52(12):2923-33.

There is also no mention of statistical significance in Figs. 5, 7, 8, and 9. Including that would yield more confidence in the results not just for the NSCS as a whole, but would reveal in what areas of the shelf and for each of the diagnostics the regression analysis is most reliable.

**Response:** We thank the reviewer for this constructive suggestion. We agree that clearly indicating the statistically robust parts of the regression patterns helps to build confidence in our results.

For the **horizontal** regression maps, we now explicitly show statistical significance where it is most critical for interpretation. In Fig. 8, each panel presents a regression-based diagnostic under different runoff and ENSO conditions. Although the exact regions that pass the 90% confidence test differ somewhat among panels, our analysis and discussion focus on the band near the 100 m isobath, where the signal is consistently robust. To avoid overloading the figure with multiple overlapping masks, we have revised Fig. 8a and 8d to include shading that marks areas below the 90% confidence level, while leaving the key region around the 100 m isobath visually clear. The same 90% threshold and testing procedure are applied consistently to other plan-view regression maps (Figs. 4 and 6).

For the **vertical** transects (Figs. 5, 7, and 9), the situation differs from the horizontal fields. Here, grid points that do not reach the 90% confidence level are almost exclusively associated with very small absolute anomalies, i.e., weak signals that are not emphasized in our physical interpretation. As illustrated by an example provided in our response as Fig. R1, such regions naturally coincide with low regression coefficients. We therefore chose not to overlay significance shading on the anomaly profiles in order to preserve visual clarity and avoid obscuring the main structures of interest (e.g., the cores of temperature, salinity, density, and stratification anomalies). In the text, we only draw conclusions from the prominent, high-amplitude features, which we have verified to exceed the 90% confidence level.

To make our treatment of significance explicit for readers, we have updated the manuscript as follows:

Lines 269–270 (Fig. 4 caption): "Shaded areas (mainly over the continental slope) in (c) and (d) denote regions where the 90% confidence level is not met; this convention is followed in subsequent regression maps."

Lines 351–352 (Fig. 6 caption): "The shaded areas indicate regions that fail to attain the 90% confidence level."

Line 421 (Fig. 8 caption): "The shaded areas denote the approximate regions that do not pass the 90% confidence test."

Line 403 (main text related to Fig. 8): "The region adjacent to the 100 m isobath consistently satisfies the 90% confidence level."

These clarifications indicate where and how the significance tests are applied, while maintaining figure readability and focusing attention on the dynamically meaningful, robust features.

[Figure]

Figure R1. Regression maps of temperature anomaly profiles ($°C$) at Transects A and B (locations shown in Fig. 4b) during the summer of positive MVPC1 years (Table 1), with the shaded area indicating where they fail the 90% significance test.

Minor point (Fig. 6 caption): I assume the shaded areas in Fig. 6c,d are those not significant at the 90% confidence level, as in Fig. 4c,d, is that correct? Please indicate here if so, of clarify if not.

**Response**: We thank the reviewer for this careful clarification request. Yes, the shaded areas in Fig. 6c,d serve the same purpose as in Fig. 4c,d, i.e., they indicate regions that do not attain the 90% confidence level in the regression analysis. We have now made this explicit in the revised caption by adding the sentence: "The shaded areas indicate regions that fail to attain the 90% confidence level." (Line 351-352).

Related to this point, what is the temporal resolution of the fields used in the MVEOF analysis? Apologies if I missed this information, but could not find it in Section 2. Figures 2-3 seem to have yearly resolution, but I assume the temporal resolution of the fields used in the regression analysis is higher. <>

**Response**: We thank the reviewer for raising this point and for carefully checking Section 2. The temporal resolution of the fields used in both the MVEOF and regression analyses is indeed annual, consistent with Figs. 2 and 3. Specifically, for each year we compute summer-mean (June–August) fields of evaporation minus precipitation (E–P), air temperature, wind stress, SST, SLA, and SSS, and then form interannual anomaly time series from these JJA means. No higher-frequency (e.g., monthly or daily) fields are used in the regression or MVEOF analyses. To clarify this in the manuscript, we have revised the description in Section 3.1 to read (lines 155–156): "…including evaporation minus precipitation (E–P), air temperature, wind stress, SST, SLA, and SSS, all aggregated at annual (June–August) resolution."

m9 (JEBAR term units) Are the quantities plotted the gradients of the baroclinic terms, or the baroclinic terms themselves? If the quantities plotted are as stated in the caption and in Equation 6, it seems the units should be m3/s2, as the lateral derivatives are not present. <>

**Response**: We thank the reviewer for carefully checking the units in Fig. 8. The quantities plotted there are indeed the baroclinic terms themselves, consistent with Equation (6), rather than their horizontal derivatives. Accordingly, their units are $m^3/s^2$, not $m^2/s^2$ as previously indicated. This has been corrected in the revised manuscript; the caption for Fig. 8 now reads: "Regression maps of horizontal components ($m^3s^{-2}$) in the JEBAR term…"

Minor edit (line 372): Cartesian coordiante -> Cartesian coordinate system

**Response**: Corrected and many thanks!

< Figure 3d. I guess the curl is the background colour and wind speed is the arrows colour. Please clarify this in the caption.>>

**Response**: We thank the reviewer for pointing this out. Yes, in Fig. 3d the sea surface wind stress curl is shown as the background shading and the wind speed as vectors. We have clarified this in the revised caption, which now states (lines 168–169): "(d) sea surface wind stress curl (N m$^{-3}$; shown by the background color) and wind speed (m s$^{-1}$; denoted by the arrows);..."

---

## Author Response (AR4)

Dear Authors.

Thank-you for your re-revised manuscript. I am asking you to please consider the "Detailed comments" below ["Technical Corrections"] and then to upload your final manuscript to the Copernicus/OS editorial system for publication. I do not need to see it again. However, it will be copy-edited and you should check that the intended meaning is retained. Please note that the Supplement will not be checked.

Thank-you for publishing in Ocean Science.

Yours sincerely

John Huthnance (editor).

**Response:** We sincerely appreciate your meticulous stewardship of our manuscript and the invaluable guidance you have provided throughout the review process. In this revision, we have carefully addressed every comment and suggestion; the manuscript has been thoroughly updated in response.

Lines 223-224. Please do not use "/" to denote alternatives. Elsewhere you use "(. . .)". [I am not fond of either but at least please be consistent.] Table 1 caption should explain the meanings of bold type in the table; the text explanation can then be omitted.

**Response:** Thanks for your notice. We have avoided using "/" to indicate alternatives in the revised manuscript, and the explanatory text for Table 1 has been relocated to the table caption. We revise the Table 1 caption as:

Line 222-225: "Table 1. The El Niño (positive MVPC1) and La Niña (negative MVPC1) years. Shelf circulation anomalies in El Niño years with low runoff and in La Niña years with high runoff are generally opposite to those observed in high runoff El Niño years; the former cases, listed as non-bold entries in Table 1, were therefore excluded from subsequent analyses"

Lines 238-240. "during . . circulation" – this clause has no verb so is unclear. ", reflecting" –> "reflect"?

**Response:** Thank you for bringing this issue to our attention. We have rewritten the sentence as follows, eliminating the subordinate clause to improve clarity:

Line 235-237: "Interannual wind anomalies further modulate this circulation: during positive MVPC1 years (El Niño conditions; see Sect. 3.1), northeastward wind stress anomalies arise (Fig. 3e) and reflect ENSO's climatic influence on NSCS circulation."

Line 298. Better with "," after ")"

**Response:** We have rewritten the sentence as follows:

Line 291-296: "It is also noteworthy from this figure that landward of the 100-m isobath (excluding the immediate nearshore), the first baroclinic Rossby radius ranges from a few to ~10 km. The radius is given by $R_o = \sqrt{g'H}/f$, where $H$ is the water depth and $f$ represents the Coriolis parameter. The reduced gravity is defined as $g' = \Delta\rho/\rho$, with $\Delta\rho$ the density difference between the upper and lower uniform layers, and $\rho$ the domain-averaged density as reference density."

Regarding equation (2). Please clarify where subscripts (x*) and (y*) are simply components and where they denote derivatives.

**Response:** In equation (2), the subscripts $(x*)$ and $(y*)$ denote the cross- and along-isobath derivatives. Only in the terms $PGF_{y*}$ and $PGF^b_{y*}$, $y*$ represents the along-isobath direction rather than derivative.

We have inserted clarifying notes at both locations in the revised manuscript.

Line 317-319: "The $x*$ is positive onshore and $y*$ is in along-isobath direction and positive $y*$ stands for the direction with deeper waters on the left-hand-side. In the terms $PGF_{y*}$ and $PGF^b_{y*}$, the subscript $y*$ represents the along-isobath direction rather than derivative."

Figures 5, 7, 9. In your response to the reviewer comment about lack of significance shading, you responded "We therefore chose not to overlay significance shading on the anomaly profiles in order to preserve visual clarity and avoid obscuring the main structures of interest (e.g., the cores of temperature, salinity, density, and stratification anomalies). In the text, we only draw conclusions from the prominent, high-amplitude features, which we have verified to exceed the 90% confidence level." I think you should please include something of this response in the text or figure 5 caption; other readers may ask the same question and should not have to "hunt" through the discussion to find an explanation. In any case, the reviewer comment will be available to readers when the manuscript is published and it could appear that you have not answered the comment.

**Response:** Thanks for reminding. We have now added the following explanation to the Fig. 5 caption:

Line 307-309: "To maintain the visual clarity of the anomaly cores, which are confirmed to

exceed the 90% confidence level, significance shading is omitted from the profiles, and this approach is also applied in Fig. 7 and Fig. 9."